# Towards A Richer 2D Understanding of Hands at Scale

**Tianyi Cheng**[*1]   **Dandan Shan**[*1]   **Ayda Sultan**[1,2]   **Richard E. L. Higgins**[1]   **David F. Fouhey**[1,3]

[1]University of Michigan     [2]Addis Ababa University     [3]New York University

{evacheng, dandans, ayhassen, relh, fouhey}@umich.edu

## Abstract

As humans, we learn a lot about how to interact with the world by observing others interacting with their hands. To help AI systems obtain a better understanding of hand interactions, we introduce a new model that produces a rich understanding of hand interaction. Our system produces a richer output than past systems at a larger scale. Our outputs include boxes and segments for hands, in-contact objects, and second objects touched by tools as well as contact and grasp type. Supporting this method are annotations of 257K images, 401K hands, 288K objects, and 19K second objects spanning four datasets. We show that our method provides rich information and performs and generalizes well.

## 1   Introduction

From the moment we wake up to the moment we go to sleep, most of us actively use our hands to shape and interact with the world. As infants, our own ability to interact with the world is accelerated and guided by our observations of others using their hands [55, 37]. These interactions involve not just the objects we directly hold, but also tools that enable us to interact with second objects, such as a spatula that enables us to push an egg in a pan. As adults, as demonstrated by the endless supply of instructional videos on the Internet [39], we can parse demonstrations of activities to help guide our own efforts; we can follow hands, the objects they hold, the grasps they use, and how they use tools to interact with the world. Accordingly, there have been many efforts on 2D hand understanding, each focusing on a different aspect. These include large-scale bounding box datasets [53], segmentation datasets [65, 3, 14], and in-lab grasp and contact efforts [50, 5]. Each contributes to the literature, but obtaining a unified understanding of hands in action, from box to segment to grasp to tool use, requires stitching together many models and datasets.

We contribute a model that predicts a rich, unified 2D output of hands interacting with objects that is effective on a variety of first and third person data. Our model predicts boxes for hands and in-contact/active objects like past efforts [53, 17], as well as *second objects*, or objects contacted by an in-use tool, which in turn enables understanding how humans use tools to manipulate their world. Like past work [53, 44], our model predicts when hands contact objects; we extend this to a richer vocabulary that distinguishes touching, holding and using, and includes grasps. Finally, by converting boxes to segments [29], our system produces segments for hands, objects, and second objects.

These outputs are built on two key components – a model and a dataset. The model is a straightforward, easily-extensible deep network for parsing interaction from an image. We build on standard detection machinery [24] with standard losses and use a simple inference method. Powering this model is a large dataset, Hands23, consisting of 257K images, 401K hands, 288K objects, and 19K second objects. Hands23 provides unified annotations for four datasets: EPIC-KITCHENS [13] VISOR [14], the 2017 train set of COCO [35], Internet Articulation [46], as well as on our introduced dataset of interaction-rich videos, New Days. All Internet data has Creative Commons licenses and has had faces obfuscated after automatic and manual detection as well as minors removed from the dataset.

---

[*]Equal contribution.

37th Conference on Neural Information Processing Systems (NeurIPS 2023).

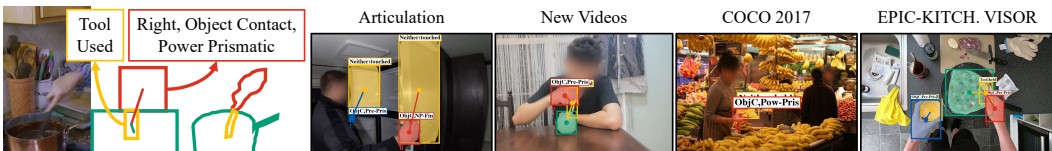

Figure 1: (Left) The rich output of our model, consisting of boxes and segments for **hand**s (**left hand blue**, **right hand red**), **objects in contact**, as well as **second objects** that are contacted by a tool. Output includes contact, making distinctions between touching and holding as well as grasp type. (Right) The method is powered by new annotations on new and existing data, covering many scenarios. Results shown are selected outputs on our test set.

We analyze our approach and dataset through a series of experiments. Across datasets, our results show that hand detection models trained on our dataset have strong zero-shot performance compared against past hand detection datasets [53, 16, 1, 44], demonstrating the expansiveness of Hands23. Within our new dataset, our experiments demonstrate that our model can detect our detailed hand-object state both well and better than past efforts such as [53]. Past work in this space [53] found adoption in a variety of communities interested in learning from hand-object interaction, ranging from parsing robotics demonstrations [31, 60, 68] to guiding video understanding [20, 15] and more [58, 40, 62]. To this end, we conclude by showing applications of our model to finding grasps in action as well as on external datasets.

## 2 Related Work

Our work aims to provide a model and dataset for obtaining rich information about hands in a wide variety of images. Our key contribution lies in enabling richer output, which includes second objects that are interacted with by tools, more fine-grained contact, grasps, and segments. As such our work spans a few directions in computer vision, machine learning, and robotics.

**2D Hand Understanding.** There have been extensive efforts to understand hands interacting with objects. One direction has been generating increasingly rich datasets for egocentric [3, 1, 12, 14, 65, 48, 21] and third person data [41, 44, 43, 53, 22, 42], with each dataset covering a particular use case and perspective. Our new dataset, Hands23 provides a unified dataset which builds on and annotates four data sources: EPIC-KITCHENS VISOR [14], COCO [35], Internet Articulation [46], and a new video dataset. Hands23 contributes to this literature on three fronts: it introduces a richer contact vocabulary than [43, 53, 21, 14] that includes fine-grained contact in terms of touching-vs-holding and tools, grasps building on the vocabulary of [11], second objects, and segments automatically obtained from SAM [29]; second, it spans both first and third person data; finally, to the best of our knowledge, it is the largest with 250K images and 400K hands. There are specialized datasets still richer than ours: [50] has a fine-grained grasp vocabulary and [65] has human-annotated segments; however, these datasets are more limited in scope and are complementary to our efforts.

To take advantage of this new data, we introduce a new detection-based model that can predict our new rich output state. We build on standard detection machinery [24] to accomplish our goal. With the advent of 2D understanding datasets, there has been a flurry of works proposing new mechanisms to recognize active objects [17, 61] and improve detection and segmentation performance [65]. We see this work as complimentary and likely to assist our model as well.

**Applications of 2D Hand Understanding.** The ability to reliably obtain basic 2D information about hands has unlocked many use cases. Naturally, some have focused on hands themselves, such as making more diverse and challenging data accessible to 3D hand reconstruction systems [7, 23, 51]. However, since hands are the key to many interactions, many other domains stand to benefit. These include ML areas like vision, where hands can guide video understanding [20, 15], as well as robotics, where hand-object detection helps parse natural human demonstrations [31, 60, 68] and video data is used for pretraining [47, 54]. Excitingly, reliable detection enables applications like post-stroke rehabilitation [58], understanding the human visual experience for neuroscience [40], and finding lost objects [62]. Our work enhances these efforts by providing richer and more accurate outputs.

Table 1: Our Hands23 and its subsets compared with the largest similar dataset, 100DOH [53]. Hands23 is $2.5\times$ larger annotated images and has 2nd objects. Each subset adds a different photographic bias [57] and challenge. New Days and VISOR [14] have many objects and 2nd objects. COCO [35] has images with many hands, but fewer second objects. Due to a focus on articulated objects in Internet Articulation [46], there are no 2nd objects.

|  | Source | #Img | >2 Hand | #Hands | w/Obj | w/2nd | #Obj | #2nd |
|---|---|---|---|---|---|---|---|---|
| 100DOH [53] | Video | 100K | 9.5% | 190K | 74.1% | 0.0% | 140K | 0K |
| Hands23 | Both | 257K | 8.6% | 401K | 71.7% | 4.9% | 288K | 19K |
| *New Days* | Video | 96K | 4.4% | 121K | 77.0% | 8.2% | 93K | 10K |
| *VISOR* [14] | Video | 38K | 0.0% | 58K | 83.8% | 10.7% | 49K | 6K |
| *COCO* [35] | Image | 45K | 33.4% | 123K | 64.4% | 1.9% | 79K | 2K |
| *Artic.* [46] | Video | 76K | 3.6% | 97K | 67.0% | 0.9% | 65K | 0K |

**Understanding Whole-Body and 3D Interactions.** Our focus is specifically hands and objects in 2D, but there is a wide literature on understanding interactions with different goals. The most related threads of research are whole body interaction and 3D hand understanding. Whole-body interactions [33, 10, 26, 56, 66, 38, 27, 45, 28] are usually evaluated on different styles of data and pose orthogonal challenges. Similarly, full 3D understanding of hands in action [64, 67, 5, 52, 42, 22, 36, 32] usually provide a richer 3D output space at the cost of working in more constrained settings. Our work may help bring these 3D techniques to more unconstrained settings.

## 3 Dataset

We contribute a new dataset of images of hands engaged in interaction with the world. All data is Creative Commons or was user-collected [12] and has had faces blurred and minors removed for privacy. The data covers four datasets: a new video dataset that we refer to as New Days, COCO [35], EPIC-KITCHENS [13] VISOR [14], and Internet Articulation videos [46]. These disparate datasets are unified by a new set of annotations including: (1) boxes for **hands**, **held objects**, and **second objects**; (2) contact information that distinguishes touching, holding, and using; (3) grasp contact labels for a subset of the data; and (4) automatically extracted segments for boxes. We include a datasheet [18] for our proposed dataset in the supplement.

### 3.1 Underlying Image Data

Our underlying data is obtained from four different datasets. These datasets cover still images, third person video, and egocentric video.

**New Video Dataset.** We gather a fresh set of video data of people engaged in interaction with the world, following the general strategy outlined in [16, 53]. The supplemental contains the full details, but in short, we follow three steps: we generate a vast pool of potential videos via a combinatorial set of text queries; we automatically select videos as likely to contain interactions (and unlikely to contain children, lecturing, or have cartoons) using a classifier trained on video thumbnails; finally, we sample frames from the selected videos to contain a mix of up-close and far-away scenes.

**Past Data.** We use three public datasets: the 2017 training set of COCO [35]; EPIC-KITCHENS VISOR [14, 13], and Internet Articulation [46]. We use the full dataset from Internet Articulation and VISOR, and use COCO images containing non-crowd people with area $\geq 10^3$ pixels.

**Data Splits.** We provide $\approx 80/10/10\%$ train/val/test splits that split video data by channel and are backwards compatible with existing datasets. These are documented in the supplement.

**Copyright & Privacy.** All underlying image data used are public; when posted by users, they were posted publicly with a Creative Commons license selected. Obtaining explicit permission from each user for inclusion in the data is impractical due to the scale of the data. However, for privacy, we have: (1) blurred faces of people via automatic methods combined with multiple rounds of manual correction; (2) found and removed images containing minors; and (3) will provide a means for people to remove their data from the dataset. These procedures are detailed in the supplemental.

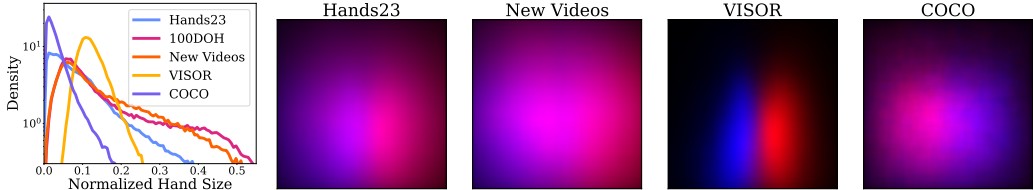

Figure 2: Dataset statistics. **(Left)** *Density* of normalized hand size (box diagonal over image diagonal) for four datasets. COCO and VISOR have a strong preferential sizes; 100DOH and New Days have wider distribution. Hands23 covers all the sizes. **(Right)** Spatial density of hand bounding boxes, showing left and right hand in blue and red respectively. VISOR has a typical egocentric side and location bias and COCO has few hands in the upper image.

## 3.2 Annotations

We provide unified annotations for all four datasets. Annotation is needed, even for a dataset like VISOR [14] that already has annotations, as VISOR "hands" have a varied definition that includes the forearm.

**Hand and Side.** All annotations are built on top of a hand. We obtain annotations for left/right hands as bounding boxes, defining hands as terminating at the wrist following [53, 43]. To avoid tiny hands in COCO [35], we only annotate non-crowd instances with at least $10^3$ pixels of area.

**Auxiliary Labels.** For each identified hand, we obtain annotations for a rich set of labels, including boxes for contacted objects and secondary objects.

*Contact.* We label contact following the contact vocabulary of [43]: {*no*, *object*, *self*, *other-person*}.

*Fine-Grained Contact: Holding-vs-Touching and Tools and Containers.* Past datasets [53, 43, 14] have *not* distinguished touching and holding objects. For more nuance, we obtain annotations for {*use tool*} and the Cartesian product of {*touch*, *hold*} and {*tool*, *container*, *neither*}. We define tools as objects that are used like a hand to physically interact (e.g., spoons, tennis rackets) and containers as objects that can hold something (e.g., bins, trays, bottles). We distinguish *use* from *hold* by requiring physical contact with a second object, and distinguish *hold* from *touch* by whether the hand is expected to drop the object. We provide these annotations only to provide potentially useful information if a downstream task aligns with these definitions. Tasks that do not distinguish tools can, for instance, marginalize over the probabilities to obtain touch/hold/use.

*In Contact Object, Secondary Objects.* For hands in contact with objects, we also obtain annotations of the box of the contacted object following [53]. For self and other-person contact, this box is the full person. Furthermore, for in-contact objects that are tools, we obtain annotations of the box of the second object (e.g., lettuce if a hand is holding a knife that is cutting lettuce).

*Hand Grasp.* For a 65K subset of hands in contact, we obtain seven grasp types. The first distinction annotated is *Prehensile* grasps (holding) and *Non-Prehensile* grasps (non-holding). Prehensile grasps are then divided into five categories from the Cutkosky [11] taxonomy, producing the Cartesian product of {*Power, Precision*} and {*Prismatic, Circular*} as well *Lateral* grasps. Non-Prehensile grasps are also divided into *Finger Only* and *Palm* grasps.

**Segmentation.** We automatically generate segments for the hands, first objects, and second objects by converting boxes to segments. We use the Segment Anything [29] Model (SAM) and prompt it with a box. For hands, we accept the SAM output unmodified, while for first objects and second object, we take the output with the hand mask subtracted.

## 3.3 Dataset Analysis

We next analyze the data. Examples of the proposed data appear throughout the paper, but we report some summary statistics, as well as a comparison with other datasets.

**Data Analysis.** We analyze the subsets of the data that have been annotated in Table 1 and compare with the most comparable dataset, 100DOH [53]. An extended table with more datasets appears in

Table 2: Zero-shot generalization to/from Hands23. Training on Hands23 produces good performance and sometimes better results compared to within-dataset performance; no other dataset produces good results on Hands23.

| | Hands23 | 100DOH [53] | VLOG [16] | Ego[2] | VGG [41] | TV [44] |
|---|---|---|---|---|---|---|
| Train/Test Same | 85.6 | 93.6 | 80.9 | 97.9 | 84.4 | 71.0 |
| 0-Shot *from* Hands23 | 85.6 | 93.6 | 85.5 | 96.4 | 64.4 | 60.9 |
| ↑ Generalization *from* Hands23 to other ↑ | | | ↓ Generalization *from* other to Hands23 ↓ | | | |
| 0-Shot *to* Hands23 | 85.6 | 78.5 | 64.9 | 44.5 | 56.0 | 51.7 |

the Supplement. Our new video dataset New Days has similar statistics to 100DOH and the full Hands23 is $2.5\times$ the size of 100DOH.

Due to photographic bias [57], each subset of Hands23 contributes a unique challenge: VISOR shows many tools in use and thus second objects, and COCO has many images with many people but few second objects (since still images rarely capture actions-in-progress). We spatially analyze the hands in Figure 2. Hands23 shows good coverage across sizes and locations. On the other hand, COCO and VISOR have a strong preferential size due to photographic biases (unlike internet datasets like 100DOH and New Days). VISOR similarly shows a strong left/right bias for locations of left/right hands, and COCO has few hands in the top of the image. Together, however, the datasets provide good coverage of a wide variety of scenarios.

**Cross-Dataset Performance.** To analyze Hands23's difficulty, we do cross-dataset analysis for hand detection. We train a ResNet-101 [25] Faster-RCNN [49] to follow [53]. We report zero-shot generalization performance compared to within-dataset train/test performance in Table 2. Training on Hands23 is often close to within-dataset performance. On the other hand, training on other datasets and testing on Hands23 produces poor performance. 100DOH [53] is the best performing dataset and produces a 7% AP drop (in addition to having a less rich set of labels).

**Demographics and Realism.** Hands23 has the demographics of its underlying data and is thus not representative of the world's people. We did an initial check for the orders of magnitude differences seen in other models [6]. We compared results across binarized perceived gender presentation and Fitzpatrick scale (1-3 vs 4-6) following [6]; we note the same caveats about binarization.

We obtained 100 samples of each category and compared hand/object false positive/negative rates using the model described in subsequent sections. VISOR was both substantially easier for detection and heavily skewed light-skinned in our analysis. Controlling for all such difficulty factors relating to demographics (e.g., video styles) is challenging, but to control for this known factor, we excluded VISOR. A full evaluation is in the supplement, but most error rates were close and within 1 standard error of the mean: e.g., hand FPRs across skin tone were $3.2 \pm 1.8$ (darker) vs $3.0 \pm 2.1$ (lighter) and across perceived gender were $4.3 \pm 2.1$ (female) vs $2.2 \pm 1.6$ (male). We did observe a slight increase in object FPR across perceived gender ($24.5 \pm 4.9$ female vs $16.9 \pm 4.3$ male) and skin tone ($16.0 \pm 3.8$ darker vs $12.1 \pm 4.0$ lighter), although neither was significant according to Fisher's Exact Test [59]. We nonetheless would recommend that users be cautious and not deploy the system in high stakes settings.

Regardless of demographics, the content of the data is not necessarily representative of everyday life. For instance, many people who are eating are doing performative or competitive eating. Thus, while the data depicts actual interactions of people with objects (e.g., how one picks up an egg), we urge caution in learning priors (e.g., will someone eat 50 hard boiled eggs).

# 4 Method

For simplicity, we build our approach on top of standard RCNN object detection machinery [49, 19]. Our goal is to provide a simple and easily extended model for the community to build upon. In addition to predicting the object class, bounding box regression, and segment from a ROI pooling layer, we also predict a set of auxiliary labels. Our experiments are done with a standard MaskRCNN [19] model, but our approach can be generalized to other frameworks like DETR [8]. Conceptually, the

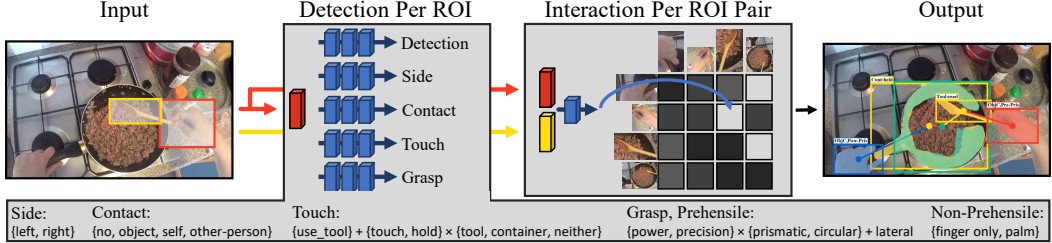

Figure 3: Our approach predicts: (1) rich information per-region of interest (ROI) that is done with MLPs on top of standard instance segmentation machinery; and (2) an interaction between pairs of ROIs (here, a hand plus a held object). The final results merge these outputs with a simple algorithm.

underlying detection architecture needs to provide a feature $\mathbf{f}_i$ for the $i$th hypothesized box. Given this feature, we predict the labels annotated in our dataset as follows (illustrated in Figure 3):

**Standard Detection.** The model predicts the standard targets, consisting of the object class and bounding box regression parameters. In our setting, in addition to the implicit background class, there are three foreground classes: **hand** (regardless of side), **object**, and **second object**.

**Segmentation.** We predict a segment using a PointRend [30] head that is set up identically to [30].

**Auxiliary Hand Information: Hand Side, Contact, Fine-Grained Contact, Grasp.** The model predicts auxiliary hand task as independent multiclass outputs using independent two-layer MLPs.

**Hand/Object, Object/Second-Object Interaction.** We train the network to classify whether *pairs* of objects are interacting. This is distinct from past work on hand-object association [53, 19] that regresses an offset vector for the hand that is compared with detected object centers.

Given features $\mathbf{f}_i$ and $\mathbf{f}_j$ for boxes $i$ and $j$, the network classifies the interaction between boxes $i$ and $j$. The network inputs the concatenation $\mathbf{f}_i$, $\mathbf{f}_j$, and a relational feature $\phi_{ij}$ and as output, predicts a multiclass output of no contact, object contact, self contact, other-person, or object-second-object. Hand/Object pairs are only trained on the first four classes; Object/Second-Object pairs are only trained on the first and last category; impossible combinations such as Hand/Hand are not trained. Because the interaction MLP is evaluated on the many *pairs* of boxes, it is a small, two layer network.

The relation feature $\phi_{ij}$ aims to help the network quickly reject unlikely interactions and includes: the vector $\mathbf{v}$ between the box centers and its normalized version $\mathbf{v}/||\mathbf{v}||_2^2$. To help recognize boxes that overlap, $\phi_{i,j}$ contains the minimum and maximum values of the signed distance function for box $i$, evaluated at box $j$'s corners, and scaled proportional to box $i$'s diagonal: these are negative if box $j$'s corners are inside box $i$ and represent the distance to box $i$ measured in hand-units.

**Inference.** For inference, we apply a simple greedy method. We start with all hand, object, and second object boxes with detection scores $\geq t_H$, $\geq t_O$ and $\geq t_S$ respectively. Then, for every hand, we find the object with the highest predicted interaction score; if the hand-object interaction score is $\geq t_A$, or the tool-interaction score is $\geq t_I$ then we accept the interaction. We associate second objects to objects using an identical algorithm and threshold. Objects without a hand, and second objects without an object are removed. We set $t_H = 0.7$, $T_O = 0.5$, $T_S = 0.3$, $T_A = 0.1$ and $T_I = 0.7$ via grid search on the validation set for thresholds that achieve best evaluation results. The performance is relatively insensitive to the hand threshold, but low thresholds for hands led to lead to poor performance on interaction prediction.

**Training and Loss Balancing.** We train using a standard Mask-RCNN recipe. All of our additional losses are standard multinomial classification problems, and so we use a standard cross-entropy loss. We sum all the losses together and weight all but one equally: we scale the association loss by 0.1, which we found to stabilize training. For the backbone, we use a standard FPN [34] ResNet101 [25] backbone. Full details are in the supplement.

## 5   Experiments

We evaluate hand, object, and second-object detection. We also evaluate our auxiliary heads, including hand side, contact, and grasp. We demonstrate results on our new Hands23 dataset, and

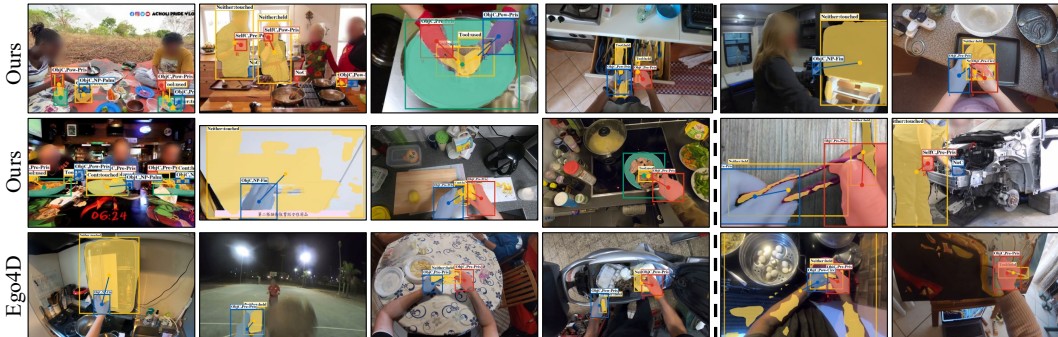

Figure 4: Results on Hands23 and zero-shot generalization to Ego4D [21] (columns: 1-4 selected; column 5,6: random). Our model performs well on different scenarios including indoor and outdoor scenes and does well on random images, including on data from a different dataset.

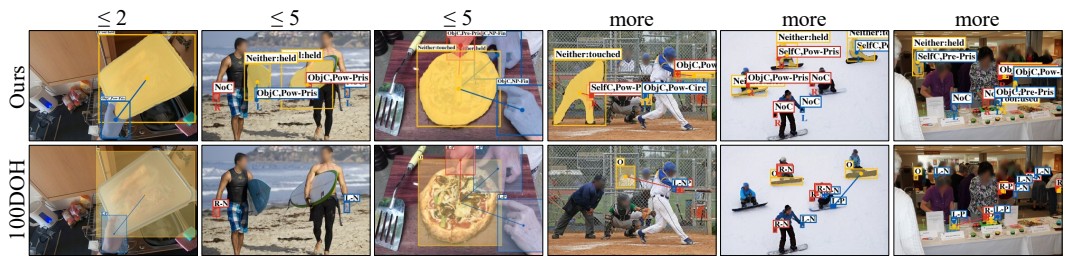

Figure 5: Comparison with the 100DOH model [53] trained on Hands23. Results are similar with few hands, but with more hands, [53] finds incorrect distant interactions. Our method is more accurate.

show applications of our method on entirely unseen data. In running these experiments, we aim to validate the usefulness of our hand-object detector to the broader community.

## 5.1 Experimental Setup

We evaluate primarily on the new Hands23, using splits defined in §3. We now define the metrics used for our output, covering box detection, segmentation, hand state and interaction classification.

*Detection.* Following past work [44, 43, 53], we use Average Precision with an IOU threshold of $0.5$. This lower threshold is used because the precise hand/forearm boundary is often unclear in images. We report the COCO AP (averaged over IOU thresholds) in the supplement.

*Segmentation.* We evaluate as in instance segmentation; for consistency, we use a threshold of $0.5$. Our ground-truth comes by prompting SAM [29] with our ground-truth boxes. We evaluated the SAM masks compared to crowd annotations. We obtained annotations for segments of 1000 hands and 1000 in-contact objects and evaluated on segments that workers came to consensus on. Median IoUs were high (hands: $0.84$, objects: $0.89$) and most objects had IoU $\geq 0.75$ (hands: $72\%$, objects: $77\%$). IoUs are close to those for COCO reported in [4]. We hypothesize that the slight drop in precision for hands is due to differences in where hands end.

*Hand State.* Past work [53] evaluated hand state questions (e.g., side) by calculating AP, but treating hands as correct only if both the box and state are correct. This metric mixes detection and state and is hard to interpret. Since hand detection performs well ($\approx 85\%$ mAP), we instead evaluate accuracy on the true positives, or the chance that correctly detected hands have the correct hand state.

*Hand Association.* Following the approach for hand state, we evaluate hand detection at a fixed operating point. For each correctly detected hand, we analyze whether there is a correctly associated object box. We define true positives as hands for which there there is an object box and the predicted object box has IOU $\geq 0.1$; we set a low IoU to account for ambiguities in the object's extent and to avoid also implicitly evaluating object detection. We define false positives as false detections, and false negatives as hands with ground-truth object boxes and no predicted boxes.

Table 3: The proposed model's performance on detection and segmentation (AP) and states (accuracy)

| Detection (AP) | | | Segmentation (AP) | | | State (Acc) | | | |
|------|------|----------|------|------|----------|------|------|------|-------|
| Hand | Obj. | 2nd Obj. | Hand | Obj. | 2nd Obj. | Side | Cont. | Fine | Grasp |
| 84.3 | 51.6 | 44.0 | 67.0 | 45.0 | 33.2 | 95.0 | 84.5 | 67.0 | 56.0 |

Table 4: Hand/Object Association Evaluation. On most data, proposed association mechanism and of [53] are similar, but [53] performs poorly with many hands. On both subsets of the data, the proposed model produces more hand detections (34.9K vs 31.6K on all, 2.3K vs 1.1K on $\geq 6$ hands).

| | (All) Prec. | Recall | F1 | ($\geq 6$ hands) Prec. | Recall | F1 |
|------|------|------|------|------|------|------|
| Proposed | 84.6 | 96.5 | 89.9 | 70.2 | 92.3 | 79.1 |
| Shan et al. [53] | 78.0 | 93.9 | 85.3 | 45.9 | 71.4 | 55.9 |

## 5.2 Results

We next analyze the performance of the system. We evaluate multiple outputs on a variety of datasets.

**Performance on Hands23.** We show qualitative results in Figure 4, including random results and quantitative performance in Table 3. More analysis appears in the supplement. Our method produces strong performance on this data: hands are detected accurately, and objects and second objects show strong performance. Performance does drop when hands and objects are small, and in cluttered images. Common grasp confusions include precision-vs-power for prismatic grasps, and precision prismatic and non-prehensile fingers. Contact is recognized well. For fine-grained contact, the most common confusions are about the object type or use vs hold: *tool held* is most confused with *tool use* and *neither held*, for instance. Interaction association performs well but struggles if there are heavy occlusions or shadows. Mask predictions do not perform well when it is a large scene object (e.g., the sea the boat is surfing on or the snow the ski pole is sticking into) and the edges are not sharp on thin or crowded objects. Common failure modes are visualized in the supplement.

**Interaction Prediction Performance.** One key methodological difference between the proposed approach and past work [53] is the mechanism for predicting interaction: [53] produces a single vector per hand that points to the center of the object box the hand is interacting with. This works well on images with relatively few hands, but does not work well on many images. We compare the approaches quantitatively in Table 4 and qualitatively in Figure 5: our approach has a slight degradation when there are many ($\geq 6$) hands while the performance of [53] plummets.

**Blurring.** We now analyze the impact of facial blurring on hand recognition by analyzing the performance of systems trained and tested with either blurred or unblurred faces. We aim to address the two questions also asked by [63]: does face blurring lead to missed detections, and does a model trained on blurred faces work when tested with unblurred faces, as will be common in practice?

We report the four options in Table 5 for detection, segmentation, and one representative hand state question, hand side. Full state is in the supplement. Working entirely with blurred data (✓/✓) causes a slight dip compared to working entirely with unblurred data (✗/✗), but performance is close. Training on blurred data and testing on unblurred data (✓/✗) is virtually identical – a tiny (0.1) drop in AP. The model has few false positives on faces, typically when they are large and in the upper half of the image. This may impact deployment and could be mitigated by either pre-blurring all input to the model or adding additional hard negatives of faces.

**Generalization to Ego4D.** Having trained an effective model, we now use it on the Ego4D [21] dataset. We show zero-shot generalization results in Figure 4. Quantitatively, our model yields a 86.9 Box AP / 60.0 Segmentation AP on the on the validation set.

We were unable to annotate Ego4D due to strict restrictions on data redistribution, but can use self-training to integrate Ego4D and our data. For each hand box, we add the output generated by our model plus SAM masks and retrain. We show qualitative results in the supplement, but quantitatively fine-tuning for 110K iterations raises Ego4D performance to 90.9 Box AP / 66.3 Segmentation AP. Continuing to train on Hands23 does not increase for the same number of iterations performance.

Table 5: Cross-facial blurring performance for hand+object detection, and hand state. For each category, we compare training with (✓) and without (✗) blurring and test in each column with (✓) and without (✗) blurring. Performance is similar. Full hand states are similar and in the supplement.

| | Hand Det. | | Obj. Det. | | 2nd Obj. Det. | | Side | |
|---|---|---|---|---|---|---|---|---|
| | ✓ | ✗ | ✓ | ✗ | ✓ | ✗ | ✓ | ✗ |
| Train ✓ | 84.3 | 84.2 | 51.6 | 51.5 | 44.0 | 42.3 | 95.0 | 95.3 |
| Train ✗ | 82.4 | 84.3 | 50.1 | 52.6 | 43.3 | 43.6 | 94.0 | 95.1 |

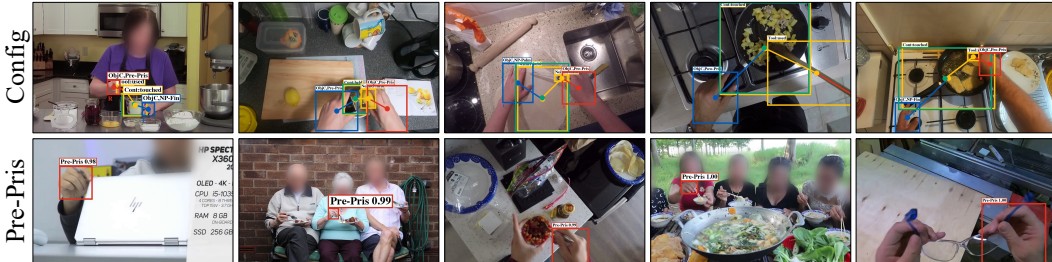

Figure 6: Applications Enabled. Top row: searching for the bi-manual manipulation *Hand → Object → 2nd Object ← Hand*. We show 5 examples selected from a set of 7 automatically found ones; the full set is in the supplement. Bottom: top precision prismatic grasps from each subset plus Ego4D.

**Applications: Finding Configurations of Objects and Grasps.** The ability to recognize grasps, hands, objects, and second objects enables us to find particular types of interactions in our data. These may help robotics research find, for instance, demonstrations of bi-manual manipulations in video data or examples of people using a precision prismatic grasp. We show some examples in Figure 6 for a particular bimanual manipulation involving a tool as well as a grasp.

## 6 Discussion, Impacts, and Limitations

We have introduced a model that produces a rich output of hands engaged with interaction, as well as a dataset for supporting this model. We hope that our model will enable autonomous systems to better understand interaction, thereby enabling applications ranging from parsing robotics demonstrations [68] to helping stroke rehabilitation [58]. Nonetheless, surveillance systems may benefit from the increased ability to recognize hands and held object. We expect that the domain shift [57] between our data and many surveillance use-cases [9] will make use less likely, but note that there are other sorts of surveillance (e.g., via interactive home systems) that may stand to benefit more directly.

Our model is imperfect. While detection often works well, small hands and hands undergoing occlusion are often missing, and occlusions and shadows interfere with interaction association. Our grasp categorization adds richness to our understanding of the diversity of hands grasps, but may fail for grasps that fall between our categories. Additionally, predictions on masks are not good when facing large scene objects and it is hard to get accurate sharp boundaries for very thin objects.

Our data is reflective of its sources, which may not be representative of the data expected in many situations. Beyond the usual vision domain shifts (the real world does *not* look like the Internet, for instance), this non-representation can be in terms of demographics. For example, since we removed minors, our model may perform poorly on children. Likewise, while we did not find orders of magnitude difference, it is likely that there are some performance differences across demographics; we thus would extensive testing before use, and discourage use in high-stakes settings. Finally, while facial blurs had limited impact on performance, they did have impact, such as on-face predictions of hands that downstream systems may need to account for.

**Acknowledgments.** This material is based upon work supported by the National Science Foundation under Grant No. 2006619. We thank the University of Michigan DCO for their tireless continued support.

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
