# Towards A Richer 2D Understanding of Hands at Scale

**Suggested reading order for reading the supplement.** The supplement is large so that it can cover all information that is needed or was promised. We have ordered our supplement in a suggested reading order that will be of the most interest for the casual reader. This is not to diminish the importance of the other sections (e.g., the datasheet or full instructions), but these are references rather than something that can be easily digested in one sitting.

**Correction to Numbers.** The *quantitative* results for a nearly identical model were inadvertently reported in two tables in the main paper. In particular, the reported numbers come from an identical model to the one described in the paper whose segments came from an internal SAM-like system (Section D) as opposed to SAM [10]. The model that is shown throughout in qualitative results and in other Tables is trained with SAM [10] masks. The two differ only in where their segmentation ground-truth come from: the architecture, code, and all other aspects are identical.

The authors regret this error and do not believe that the error alters the conclusions of the paper (indeed the models have nearly identical bounding box performance), but wish to report the correct numbers. The two corrections are as follows.

*Segmentation (Table 3).* The true quantitative performance for segmentation is ≈15% higher than were reported in Table 3 of the main paper. Our submitted numbers were taken from the model trained on our own SAM-like outputs. The segmentation performance should read as follows: Hand: 73.3 (not 55.0); Object: 51.0 (not 36.7); and Second Object: 31.9 (not 21.3). Bounding box performance is effectively identical.

*Blur-vs-unblur Experiment (Table 5).* We reported bounding box performance here, using models trained with masks from our own internal SAM-like system. as shown in Section D, bounding box performance is effectively identical (Hand AP: 83.5 vs 83.5) or within the margin of error for a random seed (object: 59.5 vs 58.6; second object: 44.4 vs 45.2).

# Contents

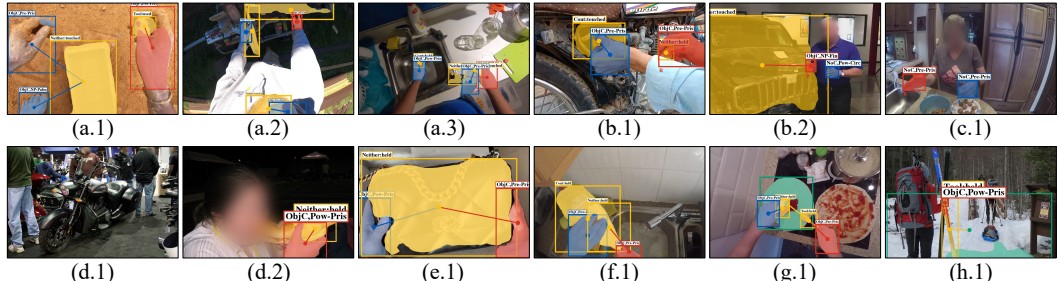

Figure 1: Failure cases. Common failure modes include false positive detection, false negative detection, missing detection, confusion between grasp predictions, confusion between fine-grained contact predictions, occlusions, and mask prediction for large scene objects.

## A   Additional Qualitative Examples

### A.1   Failure Cases

Our model is not perfect. Here we discuss common failure cases that happen during our experiments as thoroughly as possible in Fig 1 ranging from common detection difficulties to hard scenarios (occlusions, shadows, super large to tiny objects, etc.).

(a) Hand false positive detection. For example, (a.1) (a.2), and (a.3) show that foot, face, or shadow are predicted as hands.

(b) Object false positive detection. In (b.1), when the hand is curled, the model hallucinates that there is an object for the right hand of the person. In (b.2), the right hand of the person is wrongly predicted as being in contact with the car.

(c) Object false negative detection. As in (c.1) the object in the left hand is in motion and has blur and is missed by the detector.

(d) Missing hand detection. In (d.1) and (d.2), the hands are small or occluded which leads to missing detection.

(e) Confusion between Pre-Pris (Precision Prismatic) and Pow-Pris (Power Prismatic). In (e.1), the two hands are holding the object with similar grasp but are predicted differently as Pre-Pris and Pow-Pris.

(f) Confusion between Pre-Pris and NP-Finger (Non-Prehensile Fingers Only). In (f.1) the right hand is predicted as Pre-Pris but it is only contacting the bottle with fingers.

(g) Confusion between tool-held and tool-used. In (g.1) the spoon is in contact with the pizza, but is predicted as being held, as opposed to used.

(h) Bad mask prediction for large scene objects. In (h.1), the ski pole (first object) is sticking into the snow (second object). Mask prediction on such kind of large scene objects is very hard.

### A.2   Grasp Type Ranking on 4 Subsets and Ego4D

Understanding hand grasp is about understanding how hands and objects contact each other during hand-object interaction. It is critical for understanding the inter-relationship between hands and objects as well as transferring hand grasp manipulation ability to robot grasp manipulation. There are a lot of video data capturing tons and tons of hand activities, but how could we get useful information from the data?

One application of our model is to search for certain grasps in the data. Here we run our model on Hands23 testset and Ego4D valset to get all the hand grasp predictions. Ranking the grasp scores on each grasp type gives us the most typical hand grasp of each type. In Fig 2, we show the most confident sample of each grasp type on each subset of Ego4D. Since there is very little training data for Lateral grasps, no hand has lateral predicted as the most likely grasp and we therefore do not

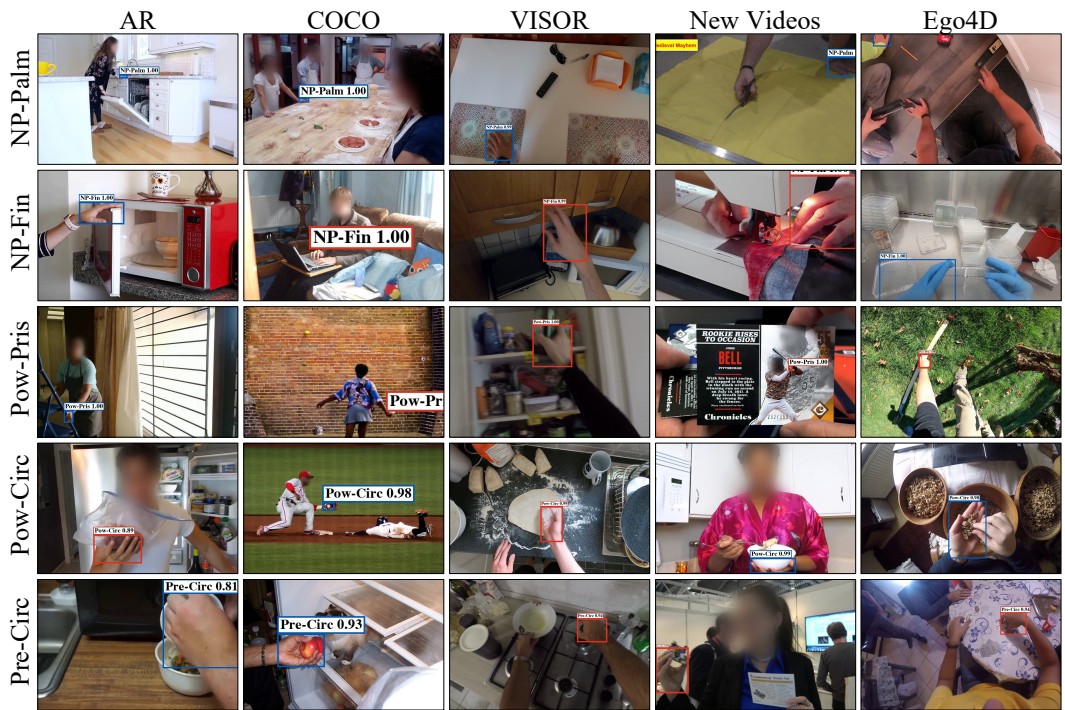

Figure 2: Grasp type ranking. Showing the Top 1 ranked sample of each grasp type on the 4 subsets (AR: Internet Articulation data [16]; COCO: COCO 2017 train [11]; VISOR [6]; and New Videos) plus Ego4D.

include it. We believe that our model is able to serve as a useful tool to collect hand grasp information on wild data, such as retrieving certain grasp types.

## A.3 More Hand Configurations

Another application is to find various hand configurations. Previously, most hand-object interaction research is focusing on one-hand-one-object interaction. However, in reality, there are more challenging hand object configurations such as bi-manual manipulation (one object interacting with two hands) and hand-tool-object interaction (the hand interacting with a tool and the tool affects the end object).

We present 6 interesting hand configurations here (although also note that there are more to explore), including the one *Hand → Object → 2nd Object ← Hand* mentioned in the paper. First, we give a description of them.

- *Hand → Object ← Hand* (HOH) is two hands interacting with the same object.

- *Hand → Object, Object ← Hand* (HO, OH) is two hands interacting with different objects.

- *Hand → Object → 2nd Object ← Hand* (HTOH) is one hand interacting with an object (tool-1) which also is interacting with a second object, while the other hand is interacting with the second object too.

- *Hand → Object → 2nd Object, Object ← Hand* (HTO, OH) is one hand interacting with an object (tool-1) which also interacting with a second object, while the other hand is interacting with another object.

- *Hand → Object → 2nd Object ← Object ← Hand* (HTOTH) is one hand interacting with an object (tool-1) which is interacting with a second object (2nd-obj-1), while the other hand is interacting with an object (tool-2) which is interacting with the same second object (2nd-obj-1).

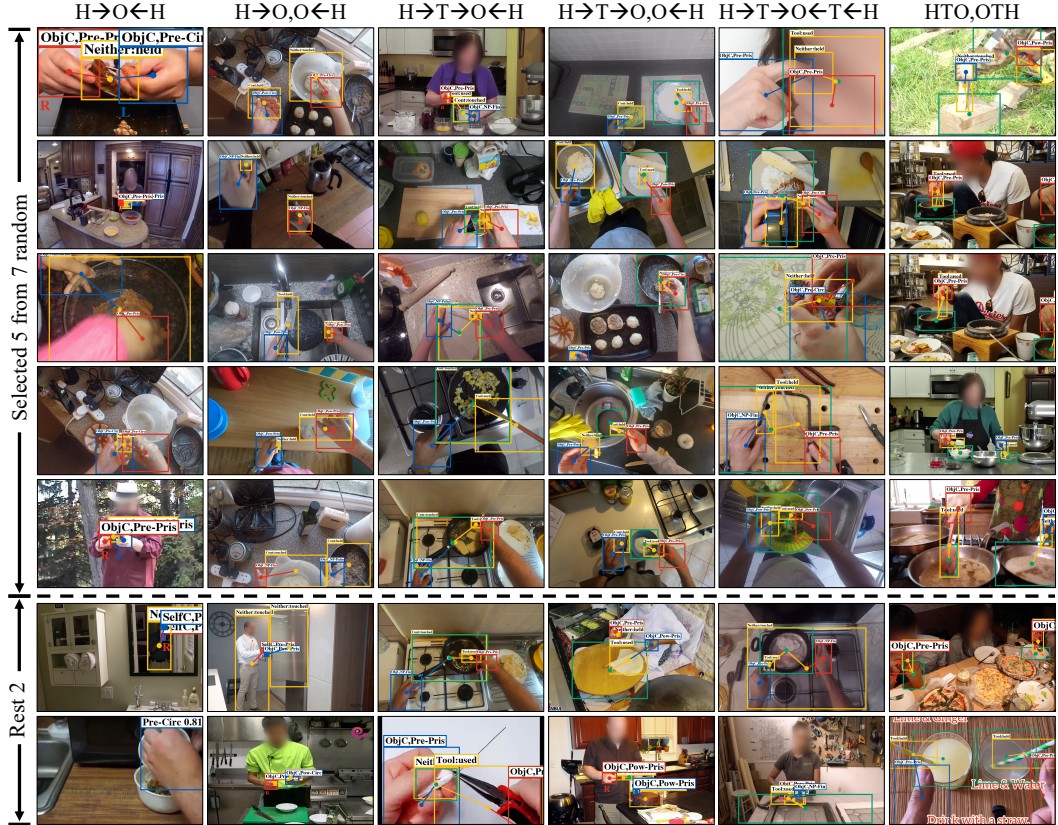

Figure 3: More hand configurations. There are various hand configurations when using hands. Here we show 6 hand configurations of bi-manual manipulations. Our model enables finding the hand-tool-object configuration.

- *Hand → Object → 2nd Object, 2nd Object ← Object ← Hand* (HTO, OTH) is one hand interacting with an object (tool-1) which is interacting with a second object (2nd-obj-1), while the other hand is interacting with an object (tool-2) which is interacting with a different second object (2nd-obj-2).

In Fig 3, we assume that the two hands belong to the same person. But as shown the in random 7 results, there are examples (e.g. the one at row 2 col 6) of the two hands belonging to two people. In the future, incorporating [14] in the searching algorithm will help associate hands with bodies and thus make sure the two hands belong to the same body. When deciding if the two objects are the same, we threshold the bounding box IoU with a relatively high value of 0.8.

## A.4 More Qualitative Examples on Hands23

In Fig 4, we provide more visualization of predictions on random images from Hands23 testset.

## A.5 More Qualitative Examples on Ego4D

In Fig 5, we provide more visualization of predictions on random images from Ego4D valset.

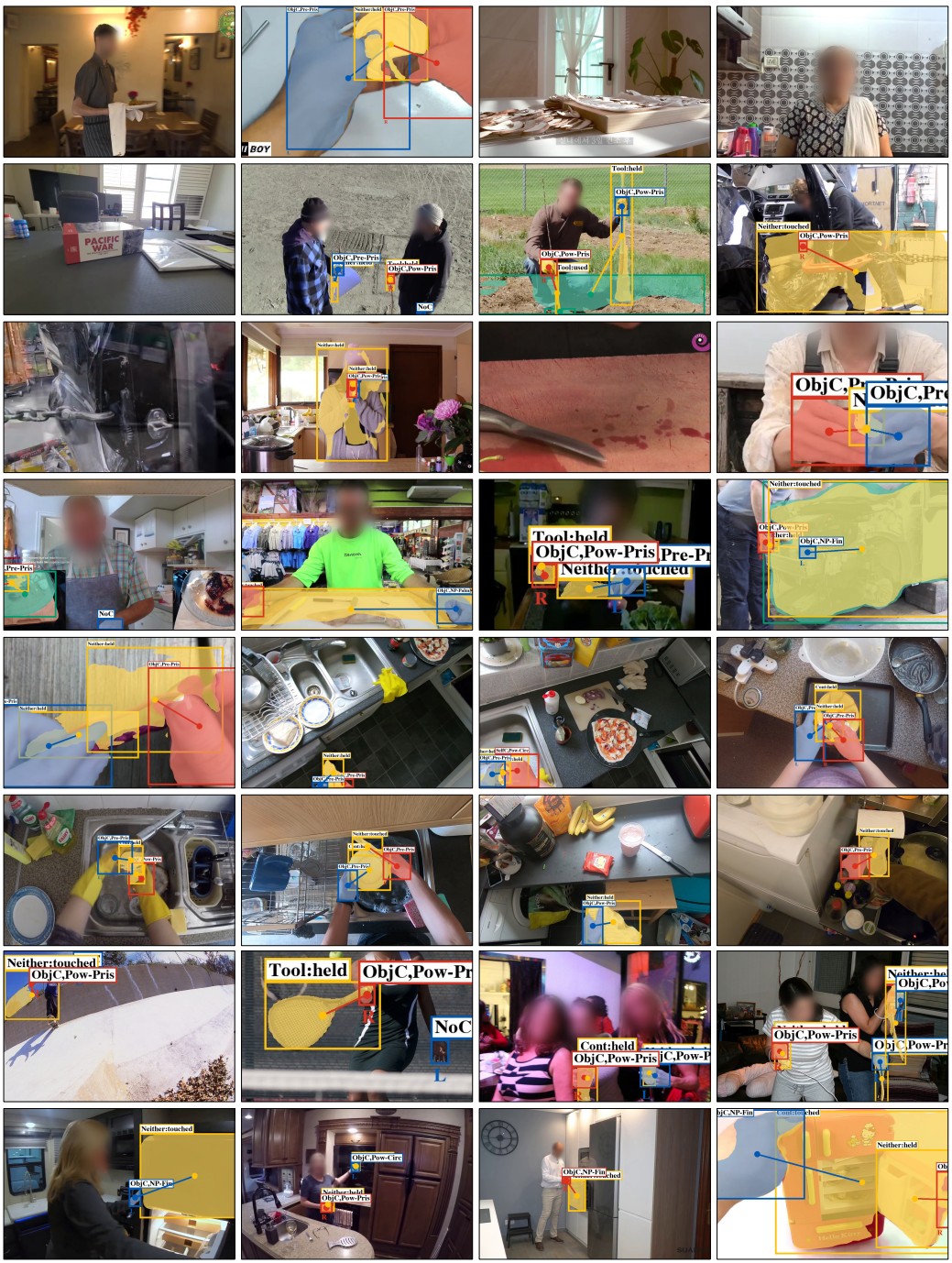

Figure 4: Random results on Hands23 testset.

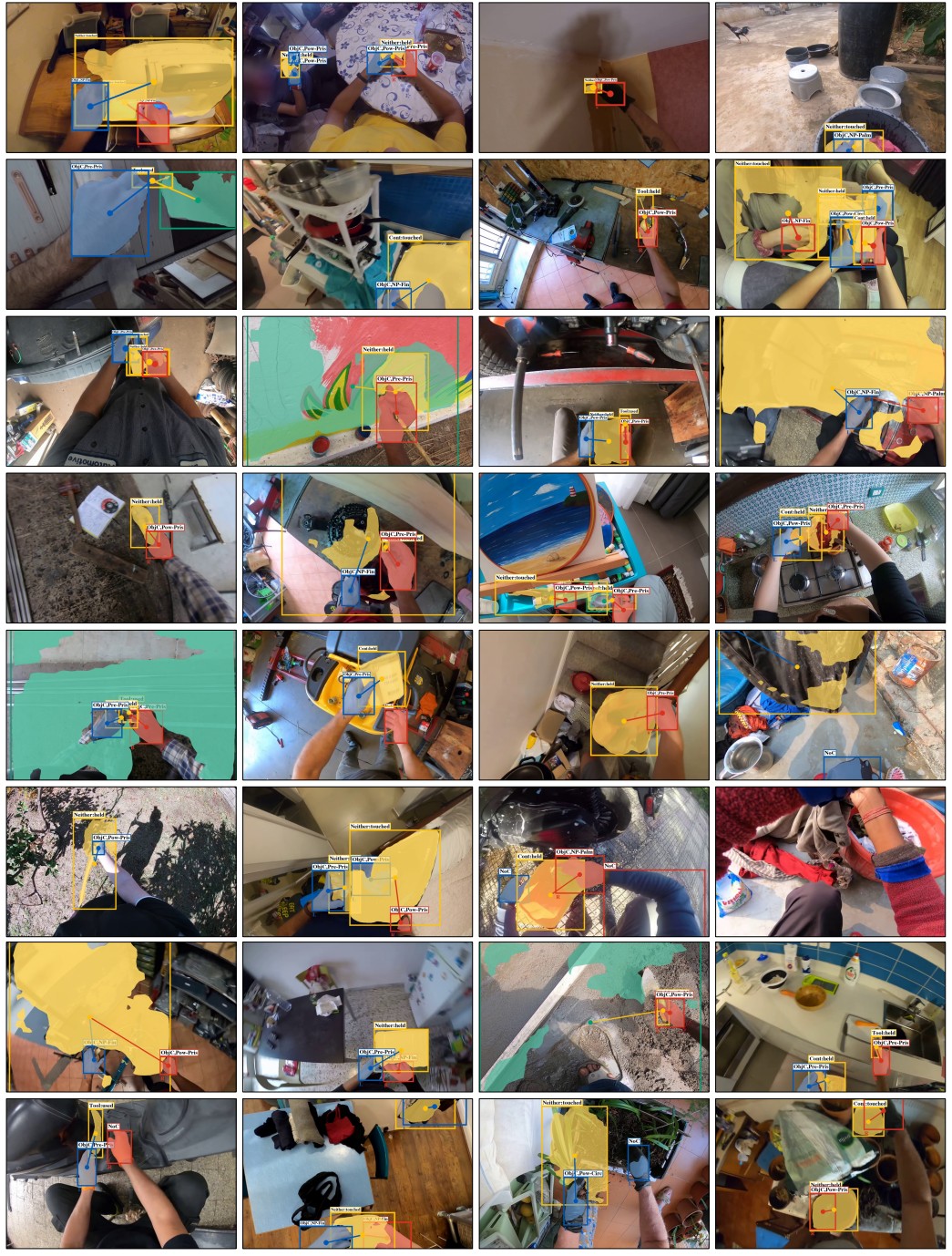

Figure 5: Random results on Ego4D valset.

Table 1: Extended dataset comparison. Compared with existing datasets, Hands23 is the first dataset that includes the second object annotations.

| | Source | #Img | >2 Hand | #Hands | w/Obj | w/2nd | #Obj | #2nd |
|---|---|---|---|---|---|---|---|---|
| 100DOH [18] | Video | 100K | 9.5% | 190K | 74.1% | 0.0% | 140K | 0K |
| Hands23 | Both | 257K | 8.6% | 401K | 71.7% | 4.9% | 288K | 19K |
| *New Videos* | Video | 96K | 4.4% | 121K | 77.0% | 8.2% | 93K | 10K |
| *VISOR* [6] | Video | 38K | 0.0% | 58K | 83.8% | 10.7% | 49K | 6K |
| *COCO* [11] | Image | 45K | 33.4% | 123K | 64.4% | 1.9% | 79K | 2K |
| *Artic.* [16] | Video | 76K | 3.6% | 97K | 67.0% | 0.9% | 65K | 0K |
| VLOG [7] | Video | 5K | 6.5% | 26.1K | - | - | - | - |
| VIVA [1] | Capture | 5.5K | 30.5% | 13.2K | - | - | - | - |
| Ego [2] | Capture | 4.8K | 73.8% | 15K | - | - | - | - |
| VGG [13] | Flickr, TV | 2.7K | 28.4% | 4.2K | - | - | - | - |
| TV-Hand [15] | TV | 9.5K | 10.7% | 8.6K | - | - | - | - |
| COCO-Hand [15] | Flickr | 26.5K | 18.4% | 45.7K | - | - | - | - |

Table 2: The proposed model's performance on detection and segmentation, comparing AP50 (commonly used in past hand detection settings) and COCO AP.

| | Detection (AP) | | | Segmentation (AP) | | |
|---|---|---|---|---|---|---|
| | Hand | Object | 2nd Object | Hand | Object | 2nd Object |
| AP50 | 83.5 | 59.5 | 44.4 | 73.3 | 51.0 | 31.9 |
| COCO | 58.4 | 35.1 | 28.5 | 54.8 | 32.4 | 15.5 |

# B  Additional Quantitative Experiments

## B.1  Extended Table of datasets.

We compare Hands23 with more existing datasets in Table 1. Hands23 is the first dataset that introduces second object annotation for understanding hand-object interaction. 100DOH also has first object annotation but the amount of first object box in Hands23 is around twice as much as that of 100DOH. VLOG has object annotation but that is clip-wise object category label instead of object bounding box label.

## B.2  COCO evaluation numbers for detection

We also report the COC mAP (averaged over IOU thresholds) of the detection performance of our model in Table 2. Performance is lower, suggesting that there is lots of room for improvement by subsequent models in precise segmentation of the objects.

## B.3  Full Blur No Blur Tables

We report the full performance for all four combinations of training/testing on blurred and non-blurred images in Table 3. Performance is largely identical. Training on unblurred data and testing on blurred data produces the worst results consistently; however, the gap is relatively small. We do observe a small number of false positives on faces when training on blurred data and testing on unblurred data.

## B.4  Audit for Differences in Performance across Skin Tone and Gender Presentation

We report performance across Female/Male and darker skin (Fitzpatrick 4 - 6) and lighter skin (Fitzpatrick 1-3). We quantify this with both the rate (i.e., number of false detections per image as a percent) and then Fisher's exact test. We test whether there is a difference in the number of images with an error per column. We obtained these results by selecting 100 images for each category, then excluding ambiguous cases and EPIC-KITCHENS due to its substantial skew in skin-tone. Two

Table 3: Complete comparison of training/testing on ✓blurred and ✗non-blurred scenes. Performance is largely the same across the conditions.

| | Detection (AP) | | | Segmentation (AP) | | | State (Acc) | | | |
|---|---|---|---|---|---|---|---|---|---|---|
| | Hand | Obj. | 2nd Obj. | Hand | Obj. | 2nd Obj. | Side | Cont. | Fine | Grasp |
| ✓Blur → ✓Blur | 83.5 | 58.6 | 45.2 | 55.0 | 36.7 | 21.3 | 95.7 | 83.7 | 63.9 | 54.1 |
| Blur → ✗Not Blur | 83.4 | 58.5 | 45.0 | 54.4 | 36.1 | 21.2 | 95.4 | 83.6 | 63.6 | 54.1 |
| ✗Not Blur → Blur | 82.5 | 57.9 | 43.9 | 54.2 | 35.9 | 20.6 | 94.8 | 83.7 | 63.0 | 53.4 |
| ✗Not Blur → ✗Not Blur | 84.4 | 59.2 | 44.3 | 54.3 | 36.1 | 20.9 | 95.6 | 83.8 | 63.7 | 53.7 |

authors independently evaluated the outputs and counted false positives/negatives; if any annotator spotted an error, it was counted as an error.

There is not the yawning gap exhibited in the GenderShades [3] paper and error rates are typically quite close. There is a slight increase in errors that is not statistically significant, especially for object FPRs. We believe that various uncontrolled statistical biases are still present in the data, for instance in terms of the subject matter and composition. However, we urge that downstream users monitor output to see if they see these performance differences play out in their own data.

Table 4: Audit of performance across skin tone and presented gender. We report (in percentage) the false positive rate and false negative rate for hands and objects. We additionally report the p-value of Fisher's exact test, testing whether there is a difference in the number of images with a false positive/negative.

| | Hand FPR | Hand FNR | Obj FPR | Obj FNR |
|---|---|---|---|---|
| Female | $4.3 \pm 2.1$ | $12.8 \pm 3.8$ | $24.5 \pm 4.9$ | $12.8 \pm 3.8$ |
| Male | $2.2 \pm 1.6$ | $12.4 \pm 3.5$ | $16.9 \pm 4.3$ | $10.1 \pm 3.2$ |
| Fisher's Exact p | 0.68 | 1.00 | 0.27 | 0.81 |
| Fitzpatrick 1-3 | $3.0 \pm 2.1$ | $13.6 \pm 4.2$ | $12.1 \pm 4.0$ | $9.1 \pm 3.5$ |
| Fitzpatrick 4-6 | $3.2 \pm 1.8$ | $12.8 \pm 3.8$ | $16.0 \pm 3.8$ | $11.7 \pm 3.9$ |
| Fisher's Exact p | 1.00 | 0.81 | 0.65 | 1.00 |

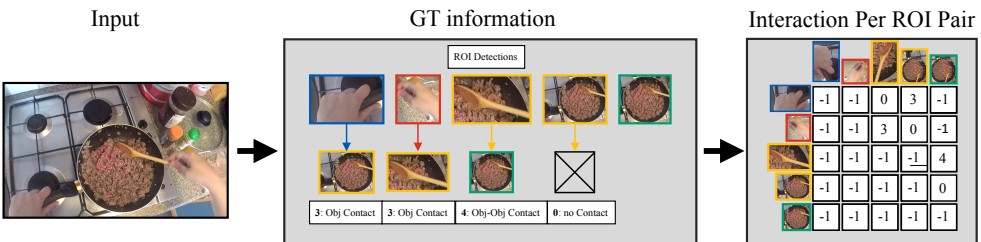

Figure 6: Illustration for interaction association table.

## C  Model Architecture and Training Details

### C.1  Model Architecture

Our model is based on Mask-RCNN [8] in Detectron2 [20] with ResNeXt [21] (X-101) backbone. We modify the data registration method and data loader to include auxiliary ground-truth labels for our method (hand side, hand contact, fine-grained object contact, grasps, and interaction association).

We overwrite the StandardROIHeads to provide auxiliary predictions. We take the object detection candidates from ROI-Pooling (suppose there are $n$ of them, each with feature size $F$) and feed them to auxiliary heads we added:

- **MLPs for Hand Side/Fine-grained Object Contact/Grasp** The $n \times F$ feature matrix was forwarded into the separate MLPs for each of the 3 tasks (two hidden layers of dim 1024 with ReLU in between). We use Cross Entropy Loss with ignore indexes for all MLPs: ignoring hand side and grasp predictions for objects and second objects (objects do not have hand characteristics); ignoring fine-grained object contact predictions for hands and second objects (fine-grained object contact is only predicted for objects).

  - **Hand Side**: ($1024 \rightarrow \text{ReLU} \rightarrow 1024 \rightarrow \text{ReLU} \rightarrow 2$)
  - **Fine-grained contact Head**: ($1024 \rightarrow \text{ReLU} \rightarrow 1024 \rightarrow \text{ReLU} \rightarrow 7$)
  - **Grasp Head**: ($1024 \rightarrow \text{ReLU} \rightarrow 1024 \rightarrow \text{ReLU} \rightarrow 8$)

- **Hand/Object, Object/Second Object Interaction** We pair $n$ detection candidates into $n \times n$ pairs for forward and loss calculation as illustrated in Fig 6. The $n \times F$ feature matrix was converted into $n \times n \times (2F + 9)$ feature matrix. This matrix captures *pairs* of object candidates, representing the pair of candidate $i$ and candidate $j$ with a $2F + 9$ feature. This feature is the concatenation of $F$ features from the candidate $i$ ROI features, plus $F$ features for the candidate $j$ ROI features, plus 9 for positional features.

  The positional features consists of: {vector $\mathbf{v}$ (2,) from the box center of object/second object to hand/object, $l_2$ norm (1,) of $\mathbf{v}$, minimum distance (2,) and maximum distance (2,) between two bounding boxes, and $\frac{\mathbf{v}}{||\mathbf{v}||_2^2}$ (2,)}.

  The $n \times n \times (2F + 9)$ is passed through a MLP consisting of two hidden layers of dimension $2F + 9$ with ReLU nonlinearities; in practice $2F + 9$ is 2057 ($1024 + 1024 + 9$). We apologize for the discrepancy with the main paper.

  The ground truth interaction table($PL$) has shape $n \times n$ where $PL_{ij}$ is:

  - ignore index        **if** pair $(i, j)$ is not valid: only pairs of hand + object and object + second object are considered. Hand/second-objects are not considered.
  - **0:** no contact        **if** $i$ is not in interaction with $j$ and could be (e.g., $i$ is a hand and $j$ is an object or if $i$ is an object and $j$ is a second object)
  - **1**: self-contact        **if** $i$ is a hand and $j$ is the person for the hand
  - **2**: other person        **if** $i$ is a hand and $j$ is another person
  - **3**: object contact        **if** $i$ is a hand and $j$ is an object
  - **4**: object/second        **if** $i$ is an object and $j$ is a second object in contact.

  This table is used to calculate the Cross-Entropy Loss (with ignored index) for interaction association.

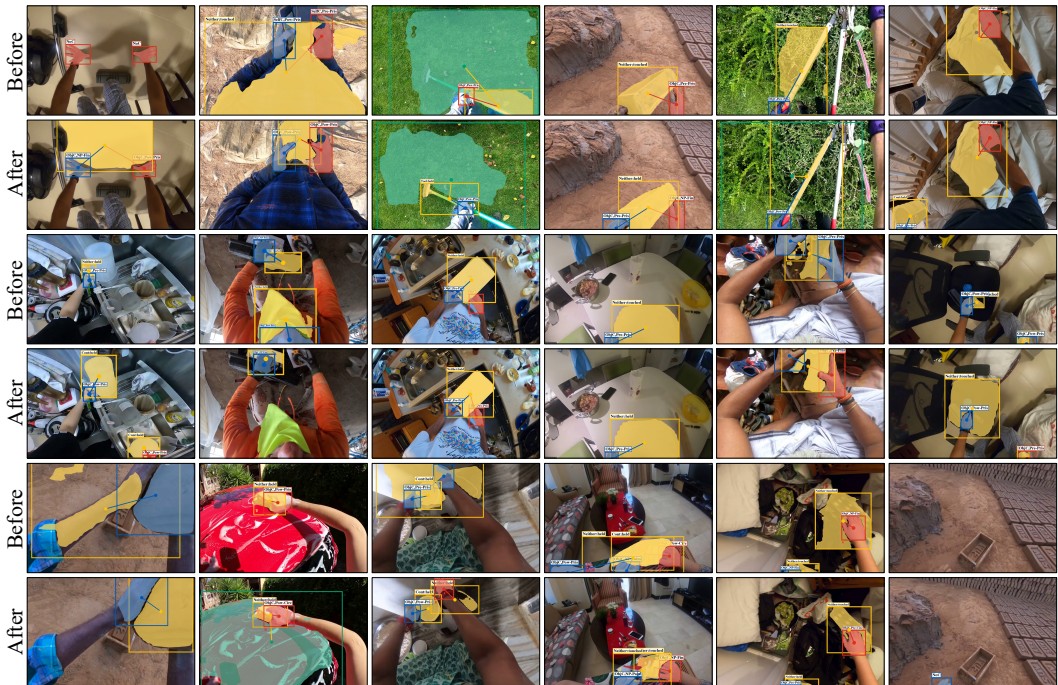

Figure 7: Performance Comparison for the model before and after finetuning on Ego4D.

Table 5: Finetuning on Ego4D. The performance on hand box, hand segmentation and hand side prediction improve after finetuning on Ego4D data. Note that Ego4D only provides side information.

|  | Detection (AP) **Hand** | Segmentation (AP) **Hand** | State (Acc) **side** |
|---|---|---|---|
| Before | 86.9 | 60.0 | 0.92 |
| After | 90.9 | 66.3 | 0.97 |

## C.2 Training Details

We have trained three different models on different versions of the data. In addition to SAM masks, we also trained our own mask segmentation models to get the automatically generated masks, which we describe in Section D. For data privacy purposes, we blurred all faces in the images. We trained models on both blurred and raw images to assess the impact of blurring.

- Model 1 (our final setting): trained on blurred images with SAM masks plus all other labels.
- Model 2: trained on blurred images with self-trained masks plus all other labels.
- Model 3: trained on raw images with self-trained masks plus all other labels.

The training recipe is the same for all models. The base learning rate is 0.01, which was scaled by 0.1 at iterations 210000 and 250000. The models were trained for 400K iterations using 8 NVIDIA GeForce GTX 1080 Ti GPUs for around 1 week.

The detection loss and segmentation loss remain the same as in Detectron2. The losses for all auxiliary heads and the interaction association are scaled by 0.1; we apologize for the incorrect scaling reported in the main paper. From our previous training experiments, training the model using learning rate of 0.01 without loss scaling on all auxiliary heads is unstable and leads to training divergence.

## C.3 Finetuning on Ego4D

We finetuned Model 2 on Ego4D training set for another 110K iterations. Comparison resutls are in Fig 7 and Table 5

Ego4D only provides hand boxes and handside labels. In addition, we provide pseudo-labels (hand contact, fine-grained object contact and hand grasp) generated automatically from our model plus SAM masks for the finetuning.

The performance before finetuning on Ego4D shows the strong generalization ability of our model. After finetuning, the performance improved. This shows that the model's performance gains with finetuning on unseen data.

Table 6: The proposed model's performance on detection and segmentation (AP), comparing using SAM [10] and our own internal system. Bounding box performance is largely identical, but segmentation is better using SAM.

| | Detection (AP) | | | Segmentation (AP) | | |
|---|---|---|---|---|---|---|
| | **Hand** | **Object** | **2nd Object** | **Hand** | **Obj.** | **2nd Object** |
| Trained on SAM [10] | 83.5 | 59.5 | 44.4 | 73.3 | 51.0 | 31.9 |
| Internal Masks | 83.5 | 58.6 | 45.2 | 55.0 | 36.7 | 21.3 |

## D  Masks from an Internal SAM-like System

The masks produced by our system are automatically generate. In the paper, we use masks that come from SAM [10]. However, during the development of the project, we had developed an in-house SAM-like system. After the release of SAM, we switched to SAM. However, we document this model and its performance as an illustration of an alternate approach, since it was inadvertently used in a few tables, and to accurately capture the compute time used in the project.

### D.1  Model Architecture and Training Details

Our model used aimed to automatically generate masks from available bounding boxes and used an HRNet [19] network with ResNet50 [9] backbone pretrained on ImageNet [17]. This model is trained on supervised examples that come from VISOR [6] and COCO [11].

To train a segmentation model in which the objective function is focused on maximizing the segmentation performance rather than localization ability, we crop and fixate hands and objects to the image center and pad to VISOR's image size to have a constant resolution for training. The same preprocessing is applied to ground truth masks so that there is a pixel-to-pixel correspondence.

**Hands.** For hands, we crop images and masks along annotated VISOR [6] bounding boxes, pad the crops to VISOR's image size and use available VISOR masks as our ground truth. After the first round of training on VISOR masks only, we use this model to generate pseudolabels for all other subsets. We then select good quality pseudolabels that cover greater than 70 percent of the bounding box area to be added to the training set in subsequent rounds. We repeat this process till the performance trajectory levels off. The final trained model is used to automatically generate hand masks for all subsets excluding VISOR.

**Objects.** Similarly for objects, we train an object segmentation model using available masks from VISOR [6] and COCO [11]. We pad all crops to VISOR's image size and objects greater or smaller than this size are either scaled up or down during inference. In this case, we do not see an increase in performance when training on additional pseudolabels, hence we halt training after the first round. This trained model is finally used to generate masks for all images with no corresponding ground truths.

### D.2  Computational Requirements

Both hand and object segmentation models were trained using a NVIDIA GeForce RTX 2080 Ti with a batch size of 1. The models were trained for a single epoch which takes about four to six days. We estimate that during development, we trained on the order of 25 versions of the model.

### D.3  Performance and Discussion

Despite the distributional shift encountered when generating masks using models trained on mostly egocentric data, we found that this approach performed quite well, with some caveats.

**Performance.** We report performance in Table 6, using SAM outputs as ground-truth. Bounding box performance is largely unaffected by changing the labels. SAM produces better segments, by about ≈15%. For metrics and a discussion of the use of SAM outputs as ground-truth, please see the main paper's metrics section.

Early in the project, when testing a model trained on only egocentric hand data, we observed that the model uses a shortcut of learning to identify skin surfaces. Since egocentric data only includes the camera wearer's hands and arms, there are no negative examples of skin surfaces the model can learn from. Based on this observation, we trained the on masks bounded by annotated hand boxes. However, this implies that the model is highly sensitive to skin tone and can only identify hands of the demographics it was trained on.

The object segmentation model performs relatively well on images where object boundaries are clearly defined and foreground-background contrast is high. We notice a dip in performance when it comes to large objects due to the limited number of large objects in the training set. Adjacently, having COCO masks in our training set significantly improved performance on very small objects.

# E   Data Processing and Redaction

We have made substantial efforts to blur all faces and remove all children from our dataset. We now describe how we did each step. All specifics about the annotation instructions appear in Section H.

## E.1   Face Blurring

We followed a process that aims to blur all the recognizable faces in the dataset. This follows a multi-step process that is partially automated but has several manual checks.

### Generating boxes and masks

**Step 1 – Initial Boxes.** Our first round boxes come from the AWS Rekognition service. This finds most of the faces in the dataset but is imperfect (hence our multiple manual steps).

**Step 2 – Verification.** We apply our face blurring algorithm below and then ask workers to check that all faces have been blurred. We ask annotators to classify images as either "all faces blurred" or "some unblurred faces". To reduce the risk of automation bias, the gold standard checks for the workers include large numbers of images for which one or more face detections have been dropped. Thus, workers see a fairly large number of images with unblurred faces.

**Step 3 – Manual Spotting.** Many of the missing faces are simply faces from unusual angles that are easily spotted by a human but understandably missed by a computer system. We ask workers to annotate these with a box, focusing on faces that are clearly visible and large enough to recognize (e.g., not 2 pixel tall faces in crowds).

**Step 4 – Verification of Manually Annotated.** We then run the images with the additional boxes through our face blurring system, and ask workers to classify each image as either "all faces blurred" or "some unblurred faces". We apply similar gold standard checks to Step 2.

**Step 5 – Manual Annotation, Including Masks.** The remaining faces are difficult to annotate and primarily depict outdoor scenes with many people. Many have a face or two missing in an otherwise properly parsed scene. Some, however, show systematic failures where large numbers of people have clearly visible faces that are large enough to be recognizable, but are entirely missed by the automatic system. We hypothesize that these are due to systematic gaps in the training data.

These images are often complex and so the authors of the paper marked these images themselves. Using photo editing programs, we marked regions to: (1) provide a bounding box for a face; (2) provide a region that needed to be blurred. The ability to blur an entire region enables us to quickly, for instance, blur a few hundred faces in tennis stands.

The boxes for redaction come from the union of Steps 1, 3, and 5. The final additional redaction mask comes from Step 5.

### Redaction Algorithm

We follow the strategy of [22], but make a few changes that catch some edge cases we observed in our data. In particular, the photos we interact with often have fairly large ranges of depths of faces.

**Input.** As input, we are given an image $\mathbf{I} \in \mathbb{R}^{H \times W \times 3}$, a set of $N$ boxes $B = \{(x_0^i, y_0^i, x_1^i, y_1^i)\}_{i=1}^N$ for the faces, and a redaction mask $\mathbf{R} \in [0, 1]^{H \times W}$ of pixels that will always be redacted.

**Data Prep.** First, we calculate a maximum diagonal across the boxes

$$d = \max_i \sqrt{((y_1^i - y_0^i)^2 + (x_1^i - x_0^i)^2)} \tag{1}$$

that defines the scale of the blur. This is used to calculate a universal blur filter for the entire image.

We then expand the boxes by a constant $c = 0.15d$, or $B' = \{(x_0^i - c, y_0^i - c, x_1^i + c, y_1^1 + c)\}_{i=1}^N$. These new boxes $B'$ define the region that will be redacted.

We then create a pixelwise "not a face" mask $\mathbf{M} \in [0,1]^{H \times W}$. $\mathbf{M}[x,y]$ is 1 if only the two things are true: (1) $(x,y)$ is outside a box in $B'$ (i.e., was not marked as a box); (2) $(x,y)$ is **not** marked as to-redact in $\mathbf{R}$ (i.e., was not marked as a redaction region).

**Blurring.** If we then define $\mathbf{G}$ as a Gaussian filter with standard deviation $0.1d$, we compute an alpha mask $\mathbf{A} = \mathbf{M} * \mathbf{G} \odot (1 - \mathbf{R})$, which smoothly blends from blurred to unblurred while also hard-forcing anything inside a redaction mask to be blurred. Then the final image is

$$\mathbf{A} \odot \mathbf{I} + (1 - \mathbf{A}) \odot (\mathbf{I} * \mathbf{G}), \tag{2}$$

which uses $\mathbf{I} * \mathbf{G}$ (the blurred image) inside boxes and $\mathbf{I}$ outside, with a smooth tradeoff (except for redaction masks, where the cutoff is sudden). When we re-save images, we use PIL's `.save(...,quality="keep")` ability to re-use the DCT coefficients to avoid double-JPG artifacts.

Our algorithm differs slightly from [22] by expanding **all** boxes by a fraction of $d$. We found that when faces varied in sizes, sometimes the redaction mask would get too blurred in $M * G$, and so high frequency details of far away faces would peer through.

### E.2 Child Detection

To mitigate concerns about the use of images containing children, even in publicly available and creative-commons data, we filtered the data to remove children. We asked workers to annotate whether any people under the age of 18 were in the picture. We only include images where annotators came to a consensus that no children were in the picture. Both images of children and images with inconsistent annotations were not included.

During the process of model development and data processing, we periodically came across pictures of children (often in large crowds); we added these to the removed list.

We initially experimented unsuccessfully with an automatic approach that used face detection and age regression. In short: we estimated the ages of people in the images based on detected faces and then removed any face that was detected as a minor. We found this approach to be too inaccurate in terms of both false positives and negatives. False positives (adults detected as children) were somewhat idiosyncratic. False negatives were primarily people who were obviously children due to context (e.g., clothing and size) but whose face were not visible or not clear.

## F New Videos

We gathered a new video dataset using a semi-automatic method. This approach combines a small amount of annotation with automatic approaches for relevance feedback. All the specifics about the annotation instructions for this task appear in Section H.

### F.1 Video Selection

We start with a collection of 9623 search terms generated combinatorially from Section F.3. For each term, we return up to 200 videos (4 pages of results with up to 50 videos per page). **We search only for videos explicitly marked as Creative Commons**. This returns 508,716 videos. We follow the approach of [7, 18] where we use YouTube thumbnails to identify videos that are likely of interest. The advantage of thumbnails is that they are substantially smaller than the video (typically under 100KB).

**Video representation for deep networks.** Given four thumbnail images from a video, we represent each thumbnail with the final feature activation of an imagenet pre-trained resnet 50. We then represent the video with an aggregation of the thumbnail representations. The video representation is the concatenation of: the mean across dimensions, L2-normalized; the standard deviation across dimensions; and the minimum, mean, and maximum of the pairwise distances between the feature vectors.

We annotate two tasks in order to filter the videos.

**Task 1 Video Validity.** The annotator identifies unaccepable videos. Each annotator sees the thumbnails montaged in a $2 \times 2$ grid. The annotator categorizes the video into one of three categories.

- *(Not Real)*: These include cartoons, animations, screen recordings, slideshows, and videogames. One or two thumbnails showing a diagram or logo (e.g., a subscription request) is acceptable; more than this makes a video fall into "Not Real".

- *(Lecturing)*: These show a person sitting in front of the camera. If two or more frames show the same person, in more or less the same posture, with the same background and talking to the camera, then this is a "Lecturing" video.

- *(Shows Children)*: If any of the thumbnails depict people who appear to be under 18, then the video is classified as "Shows Children".

- *(Acceptable)*: Anything *other* than the above is considered acceptable.

We obtain 9,856 conclusively labeled samples, of which 6570 (66.7%) are "Acceptable", 1824 (18.5%) are "Not Real", 1169 (11.8%) are "Lecturing", and 293 (3.0%) are "Shows Children".

**Video Interaction.** The annotator identifies whether the video has at least two frames of hand-object interaction. For each video, we extract its frames, and then make a $3 \times 3$ montage showing the frame at 20%, 27.5%, 35%, 42.5%, 50%, 57.5%, 65%, 72.5%, and 80% of the way through the video. Note that while the annotator sees nine frames (to see into the video to count), the network itself only has access to the four thumbnails. The idea is that the network can learn correlations between how the thumbnail is presented and the content in the video. Each annotator is asked to categorize:

- *(Interaction Rich)*: If there are two or more frames that show a hand clearly engaged in interaction (any form of contact other than resting on a table), then the video falls into this category.

- *(Not Interaction Rich)*: Any video showing fewer than two frames falls into this category, including videos with no hands visible

We obtain 6,082 conclusively labeled samples, of which 4606 (75.7%) are "Interaction Rich" and 1476 (24.3%) are "Not Interaction Rich".

**Filtering.** We fit two linear logistic regression model on the features. One predicts *Acceptable*-vs-(either *Not Real* or *Shows Children*); the other predicts *Interaction Rich*-vs-*Not Interaction Rich*. We then take $\approx 15,000$ videos from the intersection of the top 20% of the videos sorted by p(*Acceptable*) and the top 20% of the videos sorted by p(*Interaction-Rich*). We take random samples from the top instead of the top predicted to ensure that our videos are representative of "interaction-rich" videos as opposed to videos that maximally represent "interaction-rich".

### F.2 Frame Selection

Once we have selected $\approx 15,000$ videos, we extract one frame per second per video to generate a pool of potential frames.

**Frame Representation.** We represent each frame using the final feature activation of an Imagenet pre-trained Resnet-50.

**Scene Depth.** The annotator identifies the scene depth, split into three categories:

- *(Up Close)*: This frame is probably within 50cm of the camera.

- *(Further)*: This frame is probably at least 1m away. If hands are visible, they are at least 1m away.

- *(Not Real Video)*: This frame does not show a real frame (e.g., a diagram or text). We provide this as an option to ensure consensus on the handful of non-real frames that are left.

Annotators are instructed to make a best guess for videos showing scenes with depths between 50cm and 1m. All qualifiers and gold-standard tests are clearly in one category or another. We obtain 4979 conclusive samples, of which 3574 (71.7%) are *Up Close*, 1364 (27.4%) are *Further*, and 179 (3.6%) are *Not Real Video*. We fit a multinomial logistic regression model to classify each video into these categories. We then sample 50,000 frames randomly from the frames where p(*Up Close*)$> \frac{1}{2}$ and 50,000 frames randomly from the frames where p(*Further*) $> \frac{1}{2}$.

## F.3 Search Grammar

We followed the following search grammar, following [18]. The data for each of the 12 categories is generated by selecting a word from row, where $\epsilon$ is the empty string. Therefore, the DIY grammar includes the searches "DIY IKEA genius" and "furniture amazing" and "creator hacks".

Beauty:

- beauty, haircare, bodycare, make up, skincare
- routine, tips, tutorial, with me, secrets, $\epsilon$
- morning, night, anti-aging, essential, affordable, at home, everyday, natural, realistic, ultimate, winter, summer, fall, autumn, spring, 2015, 2016, 2017, 2018, 2019, 2020, 2021, 2022, $\epsilon$

Board Games:

- play, how to play, learn to play, win in
- board game, backgammon, checkers, chinese checkers, chess, darts, Go, halma, lotto, ludo, mah jongg, monopoly, pachisi, scrabble, shovel board, snakes and ladders, tic tac toe, tic-tak-toe
- 2015, 2016, 2017, 2018, 2019, 2020, 2021, 2022, game, basic, beginners, master, guide, strategy

DIY:

- DIY, $\epsilon$
- IKEA, gift, furniture, crafts, room, food, drink, decor, experiment, bag, waste, card, candy, cookie, desk, creator, boxes
- ideas, cheap, genius, master, amazing, office, home, random, hacks, 2015, 2016, 2017, 2018, 2019, 2020, 2021, 2022, $\epsilon$

Drinks:

- made, make, kitchen, home made
- sip, tea, gulp, fizz, mate, milk, gulp, draft, cider, cocoa, mixer, coffee, cooler, posset, drinks, frappe, hydromel, smoothie, syllabub, wish-wash, refresher, ice milk, milkshake, soft drink, water, espresso, cappuccino, latte
- 2015, 2016, 2017, 2018, 2019, 2020, 2021, 2022, $\epsilon$

Food:

- make, cook, cooked, restaurant, home made
- meat, comfort food, pasta, bread, yolk, chocolate, foodstuff, baked goods, junk food, loaf, seafood, beverage, slop, fare, butter, comestible, produce, leftovers, miraculous food, soul food, feed, coconut, fish, food, yogurt, breakfast food, pizza, convenience food, cheese
- kitchen, restaurant, 2015, 2016, 2017, 2018, 2019, 2020, 2021, 2022, $\epsilon$

Furniture:

- install, assembly, home
- nest, lamp, seat, table, buffet, cabinet, bedstead, etagere, washstand, bookcase, furniture, sectional, lawn furniture, chest of drawers, bedroom furniture, dining room furniture, wardrobe, $\epsilon$
- home, 2015, 2016, 2017, 2018, 2019, 2020, 2021, 2022, $\epsilon$

Gardening:

- backyard, indoor, garden, gardening, plant, grow
- care, vegetable, flower, tree, veggie, food, seed, greens, $\epsilon$
- tips, idea, guide, spring, summer, fall, autumn, 2015, 2016, 2017, 2018, 2019, 2020, 2021, 2022, $\epsilon$

Housework:

- clean, redo, housework, reorganize, decorate, tidy
- room, home bedroom, house, living room, dining room, apartment, home, $\epsilon$
- motivation, tips, extreme, with me, routine, 2015, 2016, 2017, 2018, 2019, 2020, 2021, 2022, $\epsilon$

Packing:

470     • pack, packing, unpack, unpacking, wrap, unbox
471     • clothes, luggage, suitcase, bag, gift, lunch, food, travel, box, package, trip, cruise, vacation
472     • essential, guide, tips, tricks, work, 2015, 2016, 2017, 2018, 2019, 2020, 2021, 2022, $\epsilon$

473 Puzzles:

474     • solve, play, do
475     • puzzle, jigsaw puzzle, sliding puzzle, jack puzzle, burr puzzle, lock puzzle, pyramid puzzle, ring
476       puzzle, nail puzzle, lego, magic cube, Rubik's cube
477     • beginner, impossible, 2015, 2016, 2017, 2018, 2019, 2020, 2021, 2022, $\epsilon$

478 Repair

479     • repair, fix, maintain, maintenance
480     • automobile, car, machine, trunk, mechanics, Jeep, vehicle, Ford, BMW, alternator, engine, bike,
481       motorcycle, motor, generator, computer, pc, equipment, phone, earphone, watch, bulb, eletrics, electric
482       appliance, |
483     • 2015, 2016, 2017, 2018, 2019, 2020, 2021, 2022

484 Study:

485     • study, revise
486     • with me, $\epsilon$
487     • exam, finals, midterms, midtest, dissertation, engineering, physics, history, psychology, economics,
488       exam, finals, university
489     • 2015, 2016, 2017, 2018, 2019, 2020, 2021, 2022

 # G   Datasheet for Hands23

**Preamble**

The Hands23 dataset consists of annotations on four separate data sources: (1) a New Video dataset, referred to as New Videos; (2) the 2017 training set for COCO [11]; (3) the frames from the Internet Articulation Videos [16]; and (4) the training and validation frames of the EPIC-KITCHENS [5] VISOR [6] challenge.  Answering some of the standard datasheet questions involves answering questions not just about the annotation, but also about the underlying data. Where it is relevant, we have answered the question about the underlying data as well. The answers will be as follows:

**A.** This is an answer for the dataset

*A for New Videos.* This is an answer for the New Videos subset

*A for COCO.* This is an answer for the COCO subset

*A for Articulation.* This is an answer for the articulation subset

*A for VISOR.* This is an answer for the VISOR subset

**Motivation**

**Q. For what purpose was the dataset created?** *Was there a specific task in mind?  Was there a specific gap that needed to be filled?  Please provide a description.*

**A.** This dataset, Hands23 was created to provide an improvement in the scale and quality of available datasets for understanding hands interacting with the world. Past datasets have limitations in terms of the richness of their annotation. As an ancillary benefit, many past datasets have included data that was available under unclear copyright licenses and have included minors and unblurred faces. Hands23 consists entirely of creative commons videos, blurs faces, and removes minors from the data.

**Q. Who created the dataset (e.g., which team, research group) and on behalf of which entity (e.g., company, institution, organization)?**

**A.** Cannot be answered during anonymous review but will be provided upon publication.

**Q. Who funded the creation of the dataset?** *If there is an associated grant, please provide the name of the grantor and the grant name and number.*

**A.** Cannot be answered during anonymous review but will be provided upon publication.

**Q. Any other comments?**

**A.** No

**Composition**

**Q. What do the instances that comprise the dataset represent (e.g., documents, photos, people, countries)?** *Are there multiple types of instances (e.g., movies, users, and ratings; people and interactions between them; nodes and edges)? Please provide a description.*

**A.** The dataset contains images with corresponding annotations.  The instances are hands and associated information in these images. The corresponding paper *Towards A Richer 2D Understanding of Hands at Scale* describes the annotations in more detail, but a brief description follows.

There are boxes for hands as well as additional annotation in terms of side, contact state, and (for some hands) grasp type. Hands that are in contact with objects have a box for the in-contact box; objects that are labeled as tools in use have a box for the object that tool is in contact with.

**Q. How many instances are there in total (of each type, if appropriate)?**

**A.** There are approximately 257K images containing annotations of 400K hands, 288K objects, and 19K second objects.

**Q. Does the dataset contain all possible instances or is it a sample (not necessarily random) of instances from a larger set?** *If the dataset is a sample, then what is the larger set? Is the sample representative of the larger set (e.g., geographic coverage)? If so, please describe how this representativeness was validated/verified. If it is not representative of the larger set, please describe why not (e.g., to cover a more diverse range of instances, because instances were withheld or unavailable).*

**A.** There are a few downsampling steps in the creation of the dataset. For annotations, the only downsampling done is in not labeling all hands with grasp information and not labeling far-away hands in COCO. Grasp annotation is expensive, and so a random subset of hands were annotated with grasps. Far-away hands are hard to see, and so were not annotated in COCO (specifically: only non-crowd $\geq 1000$ pixel persons had their hands annotated) The important downsampling happened with in image selection. We report what we know about each dataset below, but note that people who appear to be minors have been removed from all data.

*A for New Videos.* The data was selected from a large collection of videos described in the supplement for the paper. Once videos were selected, frames were selected randomly subject, subject to an automated balancing of estimated overall scene depth.

*A for Articulation.* Articulation data comes from videos that appears to have been selected using a procedure that appears to be similar to New Videos according to [16].

*A for VISOR.* According to its documentation, VISOR frames were sampled to be denser within actions and then further selected to have reduced blur.

*A for COCO.* COCO data was gathered using the COCO pipeline.

**Q. What data does each instance consist of?** *"Raw" data (e.g., unprocessed text or images) or features? In either case, please provide a description.*

**A.** The dataset consists of images with annotations. The core instance for the dataset is a hand in an image. This hand has:

1. a box location;
2. left-vs-right as a binary classification;
3. contact state as a multi-class classification into: {*no contact*, *self-contact*, *other-contact*, *object contact*};
4. fine-grained contact state as a multi-class classification into: { *tool-used*, *tool-held*, *tool-touched*, *container-held*, *container-touched*, *neither-held*, *neither-touched* };
5. a box for the contacted object if the hand is in contact;
6. a box for the object that the contacted object is in contact with if the contacted object is a tool;
7. grasp information, for a random subset of hands in contact with objects, which is framed as a multi-class classification problem into { *Non-Prehensile-Fingers-Only*, *Non-Prehensile-Palm*, *Power-Prismatic*, *Power-Circular*, *Precision-Prismatic*, *Precision-Circular*, *Lateral*}.

Every image has zero or more hands with these annotations.

**Q. Is there a label or target associated with each instance?** *If so, please provide a description.*

**A.** Yes. Please see the above.

**Q. Is any information missing from individual instances?** *If so, please provide a description, explaining why this information is missing (e.g., because it was unavailable). This does not include intentionally removed information, but might include, e.g., redacted text.*

**A.** Yes, for two reasons. First, only a subset of hands were labeled with grasps because grasp annotation is difficult and expensive. Second, annotators could not come to a consensus on some annotations. These are marked in the dataset as unknown.

**Q. Are relationships between individual instances made explicit (e.g., users' movie ratings, social network links)?** *If so, please describe how these relationships are made explicit.*

**A.** The hand instances are not linked to each other, but the object instances are linked to the hands, and the second objects instances are linked to the objects. The objects were linked to the hands explicitly through the annotation process: the objects are labeled as "what is the object that is in contact with this hand".

**Q. Are there recommended data splits (e.g., training, development/validation, testing)?** *If so, please provide a description of these splits, explaining the rationale behind them*

**A.** Yes, we provide recommended data splits that are chosen carefully. Our splits are chosen to minimize the chance of source contamination (e.g., data from the same channel appearing in multiple splits) and maximize the agreement with existing datasets.

We split the source datasets as follows:

*A for COCO.* We annotate the training set of COCO 2017. We assign COCO images randomly into our training set (80%) of images, validation set (10% of images), and test set (10% of images).

*A for VISOR.* VISOR has a held-out test set that we do not annotate. We follow the VISOR split as follows: we make the VISOR validation set our test set; we then randomly assign VISOR's training set into our training set (80% of images) and our validation set (20% of images)

*A for Articulation.* We try to follow the split in [16] as closely as possible. However, if we know that two frames come from the same channel, we assign them to the same split. The split promotion logic is: if the channel contains a test frame, then all the channel's frames are moved to test; if a channel contains no test frames and at least one validation frame, then all the channel's frames are moved to validation.

*A for New Videos.* We split the videos by channel, aiming to assign 80% to train, 10% to validaton, and 10% to test. In other words, all of a channel's frames are in only one split. We assign channels to the split randomly, except for videos that appear in the Articulation dataset (which are assigned according to the Articulation splits).

**Q. Are there any errors, sources of noise, or redundancies in the dataset?** *If so, please provide a description.*

**A.** There are likely incorrect annotations in the dataset, as is the case with all annotations. There are no deliberate redundancies beyond what is present when annotating frames from videos.

**Q. Is the dataset self-contained, or does it link to or otherwise rely on external resources (e.g., websites, tweets, other datasets)?** *If it links to or relies on external resources, a) are there guarantees that they will exist, and remain constant, over time; b) are there official archival versions of the complete dataset (i.e., including the external resources as they existed at the time the dataset was created); c) are there any restrictions (e.g., licenses, fees) associated with any of the external resources that might apply to a future user? Please provide descriptions of all external resources and any restrictions associated with them, as well as links or other access points, as appropriate.*

**A.** The dataset depends on a few different source datasets. We will provide an archival purpose of the dataset for non-commercial research purposes. We will not charge a fee, but users must agree to the restrictions of the underlying data. The link for download is not available at the time of submission of the paper.

**Q. Does the dataset contain data that might be considered confidential (e.g., data that is protected by legal privilege or by doctor–patient confidentiality, data that includes the content of individuals' non-public communications)?** *If so, please provide a description.*

**A.** We do not believe so and did not find any during our use of the dataset. For Internet data, the data was posted publicly by users on photo/video sharing websites, and we expect that users would have exercised precaution. For VISOR, the capture process involves the user watching and verifying their own data, so we expect that users would have also exercised caution.

**Q. Does the dataset contain data that, if viewed directly, might be offensive, insulting, threatening, or might otherwise cause anxiety?** *If so, please describe why*

**A.** We do not believe so. However, what causes anxiety will differ from person to person – if people find videos of cooking animal meat anxiety-incuding, for instance, there are videos of this in the dataset.

**Q. Does the dataset relate to people?** *If not, you may skip the remaining questions in this section.*

**A.** Yes. The dataset relates to people.

**Q. Does the dataset identify any subpopulations (e.g., by age, gender)?** *If so, please describe how these subpopulations are identified and provide a description of their respective distributions within the dataset.*

**A.** No. We do not identify demographic information of people in the dataset, except for a post-hoc audit of model performance.

**Q. Is it possible to identify individuals (i.e., one or more natural persons), either directly or indirectly (i.e., in combination with other data) from the dataset?** *If so, please describe how*

**A.** Although we have taken the steps to obfuscate faces in the data, it is certainly possible for a person with time to identify users from the data. First, the data license for our data is creative commons, which requires attribution. This attribution intrinsically may help identify users. Second, the data itself was public on a video sharing website, so we are not releasing new data. However, we believe that the face obfuscation and removal of minors from the data provides some privacy.

**Q. Does the dataset contain data that might be considered sensitive in any way (e.g., data that reveals racial or ethnic origins, sexual orientations, religious beliefs, political opinions or union memberships, or locations; financial or health data; biometric or genetic data; forms of government identification, such as social security numbers; criminal history)?** *If so, please provide a description.*

**A.** It is possible that some information can be gleaned from the videos. However, this is data that users had uploaded and therefore the amount of information that is given away is not more than what previously was published to the Internet.

**Collection Process**

**Q. How was the data associated with each instance acquired?** *Was the data directly observable (e.g., raw text, movie ratings), reported by subjects (e.g., survey responses), or indirectly inferred/derived from other data (e.g., part-of-speech tags, model-based guesses for age or language)? If data was reported by subjects or indirectly inferred/derived from other data, was the data validated/verified? If so, please describe how.*

**A.** The data is a combination of images (which were directly obtained from Internet data or existing datasets) as well as annotated labels. The labels were annotated by multiple workers using standard labeling protocols (a qualifier to verify task understanding, checks to verify correct annotations, and multiple judgments to check for annotation consensus). The resulting annotations were checked for correctness during the process by researchers on the project.

**Q. What mechanisms or procedures were used to collect the data (e.g., hardware apparatus or sensor, manual human curation, software program, software API)?** *How were these mechanisms or procedures validated?*

**A.** The annotations were obtained primarily by working with a crowdsourcing company. The precise process is documented in the supplemental materials. The images themselves were obtained as follows.

*A for New Videos.* The data was obtained with custom scripts for scanning for Creative Commons videos on YouTube.

*A for Articulation.* Unknown to us and not listed by the authors; we expect it is similar to New Videos.

*A for VISOR.* Recorded with collaboration of the depicted people, according to the datasheet for VISOR.

*A for COCO.* Unknown to us and not listed by the authors.

**Q. If the dataset is a sample from a larger set, what was the sampling strategy (e.g., deterministic, probabilistic with specific sampling probabilities)?**

**A.** The only subsampling done in annotations is in subsampling which grasps were annotated. This was done at random. The images themselves were subsampled, which we report in the question on Composition.

**Q. Who was involved in the data collection process (e.g., students, crowdworkers, contractors) and how were they compensated (e.g., how much were crowdworkers paid)?**

**A.** Data collection involved both the researchers and crowdworkers that a third party company hired.

*Researchers.* Researchers involved in the project did pilot annotations of data and the final face blurring efforts.

*Crowdworkers.* We hired a third party company to annotate the data. This company performs annotation of a set of discrete tasks (e.g., categorization, boxes, segmentation). The use of a third party intermediary makes it hard to estimate compensation, but for transparency, we report the annotation budget and breakdown into categories in the supplementary material of the paper.

**Q. Over what timeframe was the data collected?** *Does this timeframe match the creation timeframe of the data associated with the instances (e.g., recent crawl of old news articles)? If not, please describe the timeframe in which the data associated with the instances was created.*

**A.** The annotations were collected over a nearly year-long period from late June 2022 until early May 2023. The individual data for each dataset was collected:

*A for New Videos.* The data was scanned and downloaded early-to-mid-September 2022.

*A for Articulation.* Unknown and not listed by the authors.

*A for VISOR.* April 2017 through July 2020, according to the datasheet for VISOR.

*A for COCO.* Not listed, but we presume no later 2017 and likely close to 2017.

For VISOR, the collection timeframe matches the creation timeframe; for others, the collection timeframe does not match the creation timeframe. The timestamps, for instance, on the video downloads for New Videos suggest that some videos may be as old as 2010. Judging by the image content in COCO, this data was likely captured far before 2017.

**Q. Were any ethical review processes conducted (e.g., by an institutional review board)?** *If so, please provide a description of these review processes, including the outcomes, as well as a link or other access point to any supporting documentation*

**A.** For Internet data, there were no formal review processes followed because the data was pre-existing and public and did not involve interaction with the participants. VISOR is based on EPIC-KITCHENS, which involved interaction with participants. EPIC-KITCHENS was collected with University of Bristol faculty ethics approval. These application is held at the university of Bristol and the participant consent form is available here

**Q. Does the dataset relate to people?** *If not, you may skip the remaining questions in this section.*

**A.** Yes. The dataset contains people.

**Q. Did you collect the data from the individuals in question directly, or obtain it via third parties or other sources (e.g., websites)**

**A.** This depends on the source of data. New Videos and Internet Articulation [16] come from YouTube via searching for CreativeCommons-licensed data. COCO comes from similarly searching Flickr.com for CreativeCommons-licensed data. VISOR was collected directly by and with the individuals depicted.

**Q. Were the individuals in question notified about the data collection?** *If so, please describe (or show with screenshots or other information) how notice was provided, and provide a link or other access point to, or otherwise reproduce, the exact language of the notification itself.*

**A.** No for the Internet data (New Videos, Internet Articulation, COCO); yes for VISOR.

In the case of Internet data, users had posted this data publicly to websites meant for sharing photos and videos and selected a CreativeCommons license. Thus the users who captured the photos presumably knew that their data would be public, but were not explicitly informed that their data would be used for machine learning research. As a mitigation for concerns about data use, we remove minors from the dataset and blur all the faces.

In the case of VISOR, yes. Since the data was directly collected by the participants, the participants were aware of the data collection process. All participants were given the opportunity to ask questions before participating, and they could withdraw at any time without giving a reason. Participants consented to the process and watched their footage. All participants were volunteers and were not compensated.

**Q. Did the individuals in question consent to the collection and use of their data?** *If so, please describe (or show with screenshots or other information) how consent was requested and provided, and provide a link or other access point to, or otherwise reproduce, the exact language to which the individuals consented.*

**A.** Similar to the above: for Internet data, no consent was obtained but the data was previously made public and we have removed minors and blurred faces. For VISOR, the participants consented to data collection and use and reviewed their footage before its use.

**Q. If consent was obtained, were the consenting individuals provided with a mechanism to revoke their consent in the future or for certain uses?** *If so, please provide a description, as well as a link or other access point to the mechanism (if appropriate).*

**A.** For Internet data, we will provide a mechanism to remove data from the dataset for users upon release of the data. For VISOR: participants were able to withdraw from the process at any point until the data was published by DOI. At the moment, participants are unable to withdraw their data.

**Q. Has an analysis of the potential impact of the dataset and its use on data subjects (e.g., a data protection impact analysis) been conducted?** *If so, please provide a description of this analysis, including the outcomes, as well as a link or other access point to any supporting documentation.*

**A.** For Internet data, no. For VISOR, the University of Bristol faculty ethics committee reviewed the protocol, and approved the dataset. They checked any potential impact and as the data is anonymous no further actions were deemed as needed.

**Q. Any other comments?**

**A.** No

**Preprocessing/cleaning/labeling**

**Q. Was any preprocessing/cleaning/labeling of the data done (e.g., discretization or bucketing, tokenization, part-of-speech tagging, SIFT feature extraction, removal of instances, processing of missing values)?** *If so, please provide a description. If not, you may skip the remainder of the questions in this section.*

**A.** We blurred faces in all of the data except for VISOR (which has no faces). The data is otherwise unchanged (apart from basic format processing steps such as converting videos to frames).

**Q. Was the "raw" data saved in addition to the preprocessed/cleaned/labeled data (e.g., to support unanticipated future uses)?** *If so, please provide a link or other access point to the "raw" data.*

**A.** The only raw data that exists before our images are: (a) the original source videos for the video datasets; and (b) the frames with unblurred faces. We do not plan to publicly release the frames with unblurred faces.

**Q. Is the software used to preprocess/clean/label the instances available?** *If so, please provide a link or other access point.*

**A.** Not at the moment. Most of the software is one-off scripts that are not likely not of interest due to their simplicity and non-general purpose nature. However, we are happy to share the code used for blurring upon request.

**Q. Any other comments?**

**A.** No

**Uses**

**Q. Has the dataset been used for any tasks already?** *If so, please provide a description.*

**A.** Yes, the dataset has been used for hand detection, as documented in the paper. This task requires localizing hands, the objects they hold, and the objets that are being touched by tools they use. Additionally, the task requires predicting a variety of extra properties such as contact state and grasp type.

**Q. Is there a repository that links to any or all papers or systems that use the dataset?** *If so, please provide a link or other access point*

**A.** No, not at the moment.

**Q. What (other) tasks could the dataset be used for?**

**A.** We envision that the dataset could be used for a wide variety of other tasks. Earlier datasets in this area have been used for tasks such as unsupervised learning for robotics.

**Q. Is there anything about the composition of the dataset or the way it was collected and preprocessed/cleaned/labeled that might impact future uses?** *For example, is there anything that a future user might need to know to avoid uses that could result in unfair treatment of individuals or groups (e.g., stereotyping, quality of service issues) or other undesirable harms (e.g., financial harms, legal risks) If so, please provide a description. Is there anything a future user could do to mitigate these undesirable harms?*

**A.** There are a number of important considerations for using this dataset.

First, the dataset is not necessarily representative of the world's demographics: due to the collection process, our data primarily reflects the users of YouTube and Flickr, and our egocentric data mainly comes from the EPIC-KITCHENS benchmark. If the system is used in scenarios where accuracy is critical, we would urge future users to do an evaluation on their data to make sure that there are no biases in terms of performance.

Second, regardless of demographics, the dataset does not represent real-life due to the source of data. Some of this lack of realism is missing data: COCO images rarely show transitional moments when a tool is being used to interact with an object. Other lack of realism is due to realistic interactions being chained together in unrealistic ways. For instance, New Videos contains many videos of people attempting to eat enormous amounts of food. The interaction of picking up a piece of pizza may be realistic, but the number of slices of pizza may not be.

Finally, the released data and models are trained on data with blurred faces. We find that unblurred faces are occasionally seen as hands. Future users may wish to either preemptively blur faces going into the model, or suppress detections that overlap with faces.

**Q. Are there tasks for which the dataset should not be used?** *If so, please provide a description.*

**A.** VISOR requires non-commercial research use only and so the full dataset can only be used for non-commercial purposes. A commercial license for VISOR can be acquired through negotiation with the University of Bristol.

Additionally, given the unrealistic nature of some of the underlying data, we would caution drawing conclusions from the dataset in terms of frequency of events or how people interact with objects.

**Q. Any other comments?**

**A.** No

**Distribution**

**Q. Will the dataset be distributed to third parties outside of the entity (e.g., company, institution, organization) on behalf of which the dataset was created?** *If so, please provide a description.*

**A.** Yes. The dataset will be available for non-commercial purposes publicly.

**Q. How will the dataset will be distributed (e.g., tarball on website, API, GitHub)?** *Does the dataset have a digital object identifier (DOI)?*

**A.** The dataset will be released via the project website with a to-be-determined format. We will also provide a DOI.

**Q. When will the dataset be distributed?**

**A.** Not known at this point

**Q. Will the dataset be distributed under a copyright or other intellectual property (IP) license, and/or under applicable terms of use (ToU)?** *If so, please describe this license and/or ToU, and provide a link or other access point to, or otherwise reproduce, any relevant licensing terms or ToU, as well as any fees associated with these restrictions.*

**A.** The underlying data of VISOR requires this dataset to have a Creative Commons BY-NC 4.0 license, which restricts commercial use of the data.

**Q. Have any third parties imposed IP-based or other restrictions on the data associated with the instances?** *If so, please describe these restrictions, and provide a link or other access point to, or otherwise reproduce, any relevant licensing terms, as well as any fees associated with these restrictions.*

**A.** There are no restrictions from third parties on the dataset.

**Q. Do any export controls or other regulatory restrictions apply to the dataset or to individual instances?** *If so, please describe these restrictions, and provide a link or other access point to, or otherwise reproduce, any supporting documentation.*

**A.** No. There are no restrictions beyond following the underling licenses of the image datasets

**Q. Any other comments?**

**A.** No

**Maintenance**

**Q. Who will be supporting/hosting/maintaining the dataset?**

**A.** The dataset will be released via a scheme that enables long-term preservation of the data even if there are personnel changes. The precise details cannot be revealed at the moment to preserve the anonymity of the work.

**Q. How can the owner/curator/manager of the dataset be contacted (e.g., email address)?**

**A.** The creators of the dataset will be listed in the corresponding paper and can be contacted via email once their identities are made public.

**Q. Is there an erratum?** *If so, please provide a link or other access point.*

**A.** Not yet. If there are errata or updates, we will provide them on the dataset website once released.

**Q. Will the dataset be updated (e.g., to correct labeling errors, add new instances, delete instances)?** *If so, please describe how often, by whom, and how updates will be communicated to users (e.g., mailing list, GitHub)?*

**A.** We do not have concrete plans as of yet; we will announce any updates on the dataset website once released.

**Q. If the dataset relates to people, are there applicable limits on the retention of the data associated with the instances (e.g., were individuals in question told that their data would be retained for a fixed period of time and then deleted)?** *If so, please describe these limits and explain how they will be enforced*

**A.** There are no limits on the retention of data at this point. We will monitor best practices and re-assess after one year.

**Q. Will older versions of the dataset continue to be supported/hosted/maintained?** *If so, please describe how. If not, please describe how its obsolescence will be communicated to users.*

**A.** For some changes yes. If we provide corrections to annotations or other updates that are not intended to be removing data, we will have version control. If we remove data (e.g., due to offensive imagery discovered), we will not provide public access to older versions.

**Q. If others want to extend/augment/build on/contribute to the dataset, is there a mechanism for them to do so?** *If so, please provide a description. Will these contributions be validated/verified? If so, please describe how. If not, why not? Is there a process for communicating/distributing these contributions to other users? If so, please provide a description.*

**A.** Users are free to extend the dataset on their own and create derivative works, so long as they follow the license agreements of the data. There is no official mechanism to incorporate new contributions, but we encourage others to email us to let us know about extensions and modifications

**Q. Any other comments?**

**A.** No

# H    Data Annotation and Instructions

We now describe the data annotation process. We used an annotation company, HIVE (known as *thehive.ai*) for nearly all annotation except some complex tasks that were done by the authors.

**Quality Control.** HIVE implements standard quality control mechanisms during the annotation process. These consist of qualifiers (tests during the instructions to ensure that the instructions are understood), gold standard checks (tests during annotation to ensure annotation quality), and consensus labeling of judgments. Gold standard checks were selected specifically to be non-tricky judgments: the goal was to serve as a sentinel to catch random guessing.

While the platform does not permit two-way interaction with annotators, we carefully monitored the annotators to identify if our instructions were unclear, tasks were unfair, or if there were other issues. We did this by monitoring performance on qualifier and gold-standard checks to find and remove ambiguous annotations. We also reviewed the free-form feedback and ratings of our tasks by the annotators. These free-form annotations often described the clarity of the instructions, whether they thought that compensation was in line with their expectations, and difficulty.

**Compensation and Annotator Backgrounds.** The overall annotation budget of the project was approximately $40,000. Due to the use of an intermediary, converting our spent dollars into hourly rates is difficult. However, in this section, we aim to provide as much transparency about how much was spent and on what, including detailed information about the cost of each subtask.

Due to the nature of the platform, we do not have information the location or demographics of the annotators. However, given that our tasks are primarily questions that are concretely defined in terms of physical properties, we do not expect that annotator demographics will have a large impact.

**Instruction Screenshots.** We include annotation instruction screenshots. These have also had their faces blurred and children redacted for consistency with the paper, but the annotators saw the unredacted picture.

Naturally, it is difficult to show instructions for spotting unblurred faces or spotting children when the faces in this document are blurred and children are removed. When there are faces that were *not blurred* to illustrate unblurred faces for annotators, they are indicated with a black and white checkerboard pattern. When children are removed in this section, we hide it with a black mask. One face is left unredacted to illustrate the instructions. This face belongs to Nicolas Cage, who is a celebrity.

### H.1 Hand Detection

Approximately $8,400 was spent on 280K box labeling tasks for hand detection. The particular strategy depended per dataset. We used one strategy for VISOR, another for Articulation and New Videos, and finally a third strategy for COCO. In both cases, annotators marked boxes and indicated the side (left-vs-right) at the same time.

We examined the individual results and found that some hands were correct but had only one annotator marking them. We later obtained contact state for these hands using a task that also had a not a hand option. Hands marked with a valid contact state were kept; hands marked with "not a hand" were discarded. This provided an additional set of boxes that let us reach our final number

### H.1.1 Annotation for VISOR

For VISOR data, we simply ask workers to bound hands. There is little ambiguity.

Please draw bounding boxes around hands in the image, indicating left vs right for the hand.

We will show you two images stacked on top of each other.

- The top image will indicate which people we are interested in. **Please do not annotate on the top image.** We are providing it to make it easier to see the image when you annotate.
- The bottom image is where you should annotate. **Please do annotate on the top image**

Here's how to annotate:

- We are only interested in a some people in the image. We've outlined them in red and blue in the top. Please annotate hands **only** if they belong to a person that is outlined in the top of the image
- Please draw the tightest bounding box that includes all of the hand pixels, including the wrists
- If some parts of the hand are not visible, then please draw the **smallest** bounding box that encloses all visible parts of the hand. **Please do not make any predictions about the non-visible parts.**

### H.1.2 Annotation for Articulation, New Videos

For egocentric data, we also reminded workers that left and right are mirrored depending on the camera view

Draw draw bounding boxes around hands in the image, indicating left vs right.

- Please draw the tightest bounding box that includes all of the hand pixels, including the wrists
- If some parts of the hand are not visible, then please draw the **smallest** bounding box that encloses all visible parts of the hand.
- **Please do not make any predictions about the non-visible parts.**
- Please be careful about left vs right: some of the cameras are on the person's body (first person view) and some of the cameras are pointing towards the person's body (third person view)

Here are three examples.

Example 1:

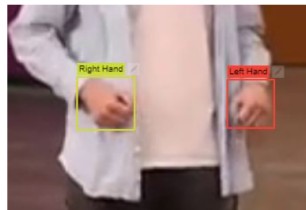

Example 2: Note that this picture is taken from the point of the view of the person who's hand this is.

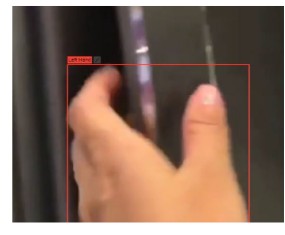

Example 3:

Example 4: Note that while the object hides part of the hand, you should annotate the full hand with a single box.

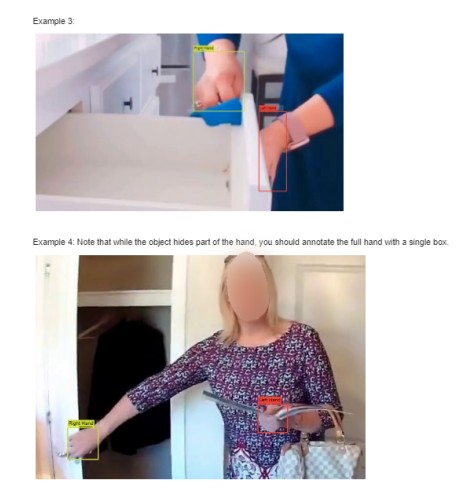

### H.1.3 Annotation for COCO

For COCO, workers annotated only hands for humans who were non-crowd and had at least 1000 pixels in area. We indicated these humans with a red and blue border as shown below. Workers were paid more for for images with more people; this rate was adjusted mid-project.

Please draw bounding boxes around hands in the image, indicating left vs right for the hand.

We will show you two images stacked on top of each other.

- The top image will indicate which people we are interested in. **Please do not annotate on the top image.** We are providing it to make it easier to see the image when you annotate.
- The bottom image is where you should annotate. **Please annotate on the bottom image.**

Here's how to annotate:

- We are only interested in a some people in the image. We've outlined them in red and blue in the top image. Please annotate hands **only** if they belong to a person that is outlined in the top of the image. Again, please annotate in the bottom image.
- Please draw the tightest bounding box that includes all of the hand pixels, including the wrists
- If some parts of the hand are not visible, then please draw the **smallest** bounding box that encloses all visible parts of the hand. **Please do not make any predictions about the non-visible parts.** The only exception to this rule are things like gloves and mittens where you know precisely where the hand is.

For example, in this below image:

- Please do not annotate the top image. This is just so we can indicate which people we would like annotated
- Please do not annotate the people who are far away. We have not outlined them in red and blue.

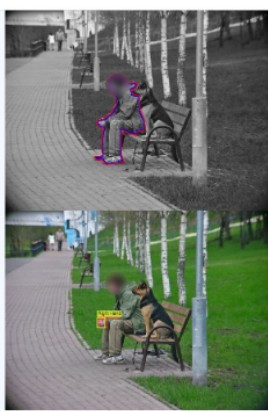

## H.2 Hand Contact State

Approximately \$5,800 was spent on 500K hand contact tasks. This was framed as a standard classification task between no contact, self-contact, other person contact, and object contact.

### H.2.1 VISOR, Articulation, New Videos

We had relatively simple annotations for these examples.

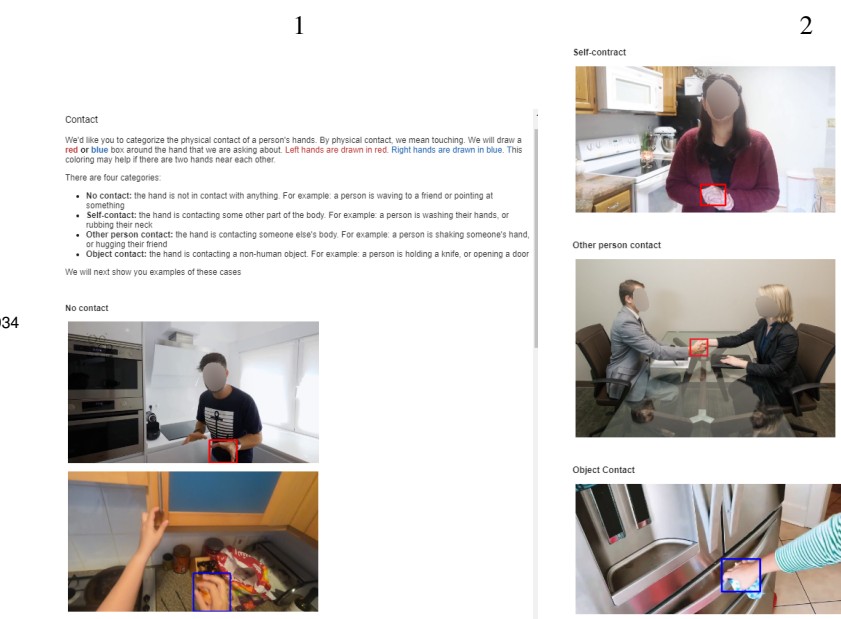

### H.2.2 COCO

We provided more examples for COCO.

1

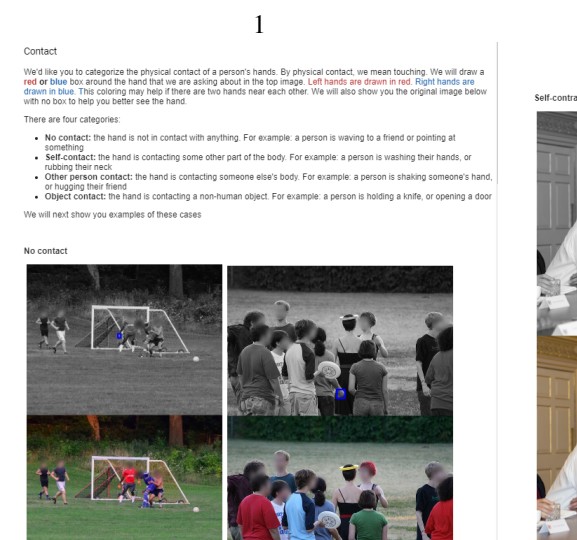

2

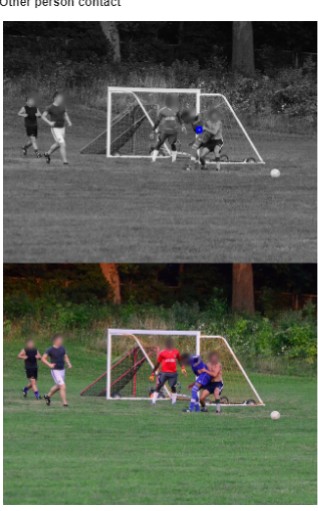

937

3

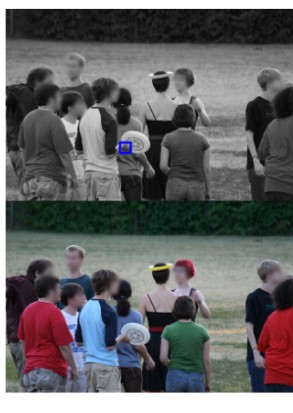

4

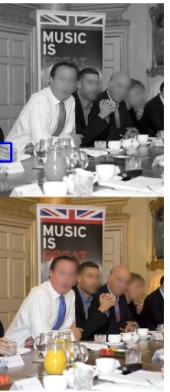

 **H.3 Additional Annotations – Checking Hands Labeled by One Annotator**

We ran an additional annotation for hands that were annotated in the box detection stage by only one annotator. We asked for the contact information or for annotators to indicate that no hand was present. Hands that were annotated with a contact state were kept as hands. The information for the additional possible label is shown below:

**Not a hand**

This box is not around a hand. Please mark incorrect boxes as "not a hand".

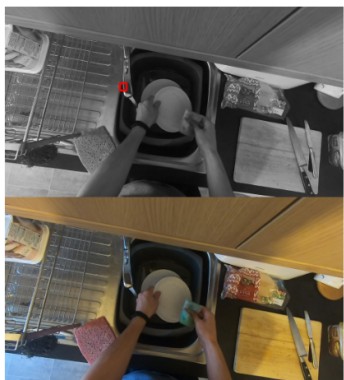

943

## H.4 Object Box

Approximately $9,300 was spent on obtaining boxes for held objects over 450K box labeling tasks. This was framed as a standard box annotation task where a single box was to be provided. Here, we provided an indication of which hand we were talking about in the top half, and asked them to annotate in the bottom half. Workers could indicate there was no box to bound.

Object bounding box

Draw bounding box around the object that is in contact with the target hand.

Each image shows a top half and the bottom half.

On the top half, we have drawn a bounding box around the target hand. **Please do not draw anything on the top half.** This half of the image is just for us to tell you which hand is the target hand. **Left hands will be drawn in red** and **right hands will be drawn in blue**.

On the bottom half, please draw a bounding box around the object that is in contact with the target hand.

You will encounter 3 scenarios, and the details will be explained in the next 3 slides.

Do not annotate on the top half. Annotate on the bottom half.

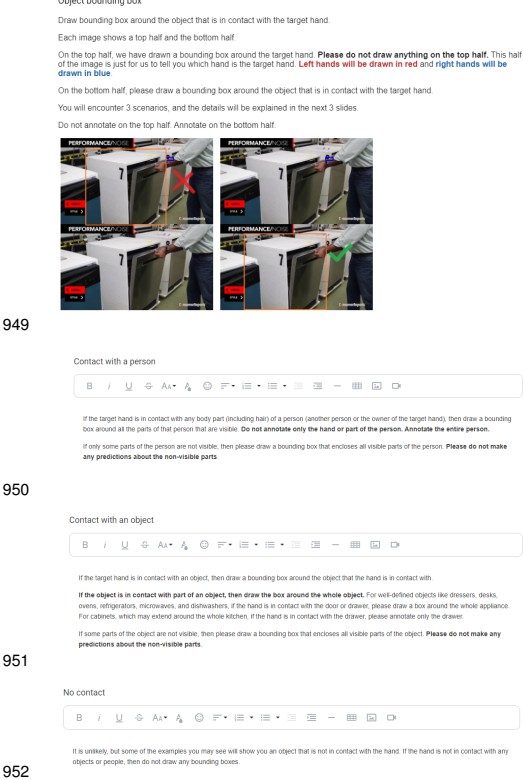

Contact with a person

If the target hand is in contact with any body part (including hair) of a person (another person or the owner of the target hand), then draw a bounding box around all the parts of that person that are visible. **Do not annotate only the hand or part of the person. Annotate the entire person.**

If only some parts of the person are not visible, then please draw a bounding box that encloses all visible parts of the person. **Please do not make any predictions about the non-visible parts.**

Contact with an object

If the target hand is in contact with an object, then draw a bounding box around the object that the hand is in contact with.

**If the object is in contact with part of an object, then draw the box around the whole object.** For well-defined objects like dressers, desks, ovens, refrigerators, microwaves, and dishwashers, if the hand is in contact with the door or drawer, please draw a box around the whole appliance. For cabinets, which may extend around the whole kitchen, if the hand is in contact with the drawer, please annotate only the drawer.

If some parts of the object are not visible, then please draw a bounding box that encloses all visible parts of the object. **Please do not make any predictions about the non-visible parts.**

No contact

It is unlikely, but some of the examples you may see will show you an object that is not in contact with the hand. If the hand is not in contact with any objects or people, then do not draw any bounding boxes.

## H.5   Object Tool/Container Status

Approximately \$5,700 was spent obtaining tool and container status over 310K classification tasks. This was framed as a classification task.

**Summary.**

Overview

Welcome!

We are trying to classify the objects people are holding in videos (with the object colored in red) and how it's being held by a hand (with the object colored in blue). We are classifying the hand and the object that is being held according to two properties.

1. **Type: what type is the object?** There are three types: *tools*, *containers*, and *neither* (not a tool or container). Tools can be used to interact physically with something (ex: a knife, spoon) and containers can contain something (ex: or a bowl, bag, or mug). Objects like laptops, bricks, tables, chairs, and cell phones aren't tools or containers.
2. **Use: what is the object being used for?** There are three options: *used*, *held/carried*, and *touched*. All types of objects can be held/carried (ex: a knife, bowl, or brick in someone's hand) or touched (ex: someone with their hands on a knife, bowl, or brick). Tools can also be *used*, or be in active use (ex: a knife cutting a potato).

Together this gives seven categories: three for tools, two for containers, and two for neither.

When we are asking the question about the blue hand and the red object. For instance, in this example, the container is being **held** by the **hand in blue**, but just touched by the hand that's not in blue.

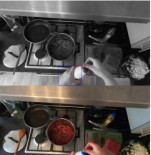

**Tools Section.**

|  1  |  2  |
|-----|-----|

Tools

Tools are objects people can use to do things in the physical world. We are interested in physical tools that can be used to physically act like a hand and are not containers.

- Examples of tools: knives, spoons, screwdrivers, ladles, baseball bats, tennis rackets, paddles are tools.
- Examples of things that are clearly not tools: bricks, tables, beds, pomegranates, flowers.
- Examples of things that are not tools in our definitions: bowls (since it's a container), cell phones, light switches, game console controllers, cameras (since they don't physically enable you to use the tool like a hand), umbrellas, clothing, or hats (since they aren't used like a hand).

For the tools, there are three states we are interested in

**1: Tool, Used**

If the tool is in contact with another object, it's in use. For instance, this spoon is in contact with the food in the pot, so it's a tool in use.

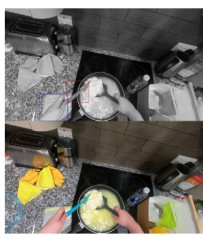

Tools

Tools are objects people can use to do things in the physical world. We are interested in physical tools that can be used to physically act like a hand and are not containers.

- Examples of tools: knives, spoons, screwdrivers, ladles, baseball bats, tennis rackets, paddles are tools.
- Examples of things that are clearly not tools: bricks, tables, beds, pomegranates, flowers.
- Examples of things that are not tools in our definitions: bowls (since it's a container), cell phones, light switches, game console controllers, cameras (since they don't physically enable you to use the tool like a hand), umbrellas, clothing, or hats (since they aren't used like a hand).

For the tools, there are three states we are interested in

**1: Tool, Used**

If the tool is in contact with another object, it's in use. For instance, this spoon is in contact with the food in the pot, so it's a tool in use.

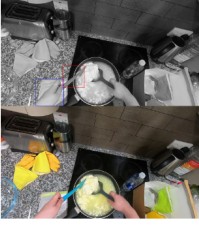

**Containers Section.**

|  1  |  2  |
|-----|-----|

Containers

Containers are object that can contain other objects. For instance, a bowl, plate, tray, bag, bottle and bin can contain other objects. For containers, there are also two states we are interested in: held vs not held. ***Please note that the qualifier for this section will contain one tool.***

4: Container, Held/Carried

If a container is held/carried it's held/carried. This is true regardless of whether it's in use or empty. For instance, all four are held/carried

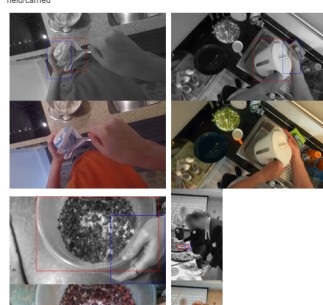

5: Container, Touched

If a container is not actually held, but just incidentally touched, then it's touched. For instance, the person isn't actually holding the container but is just moving it around by touching it.

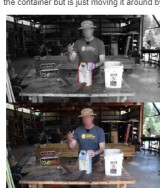

**Neither Section.**

Neither Tool nor Container

Some objects can't be used as a tool or a container. For instance, bananas, bricks, tables, and chairs.

6: Neither, Held

If the non-tool/container object is held, then it's Neither, Held

963

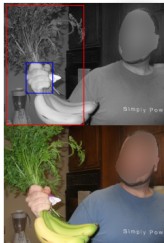

7: Neither, Touched

If the object is just touched, but not held, then it's neither, touched.

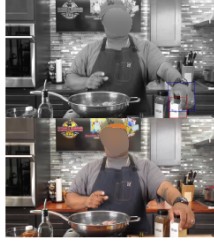
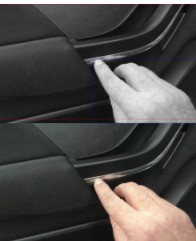

## H.6 Second Box

Approximately $1,200 was spent on obtaining boxes for a second object over 46K box tasks. This was done similarly to annotating the in-contact box: we provided the hand and object in the top half of the image and annotators annotated the bottom half of the image.

1

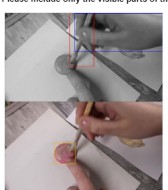

2

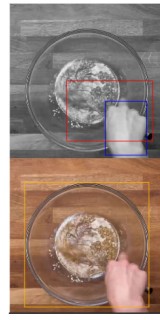
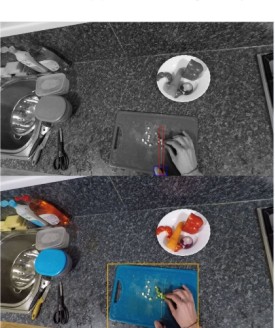

3

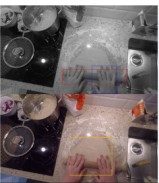

4

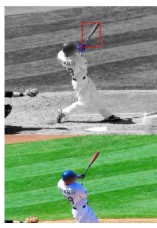
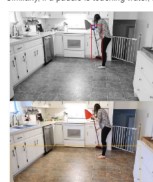

## H.7 Grasp

Approximately $3,400 was spent obtaining annotations for grasps over 154K classification tsaks. We did a pilot study of thousands of grasp annotations ourselves. This helped us identify a taxonomy that was easily annotated. We then obtained annotations hierarchically.

First, grasps were classified into NP-Palm/NP-Fin/Prehensile. Then prehensile grasps were classified into the categories described in Cutkosky [4].

## H.8 Prehensile-vs-Non-Prehensile Grasps

In our first round, we obtain annotations of prehensile grasps (where the object is held) compared with two types of non-prehensile grasps: where fingers make contact and where more than the fingers make contact. We frame this as a classification problem.

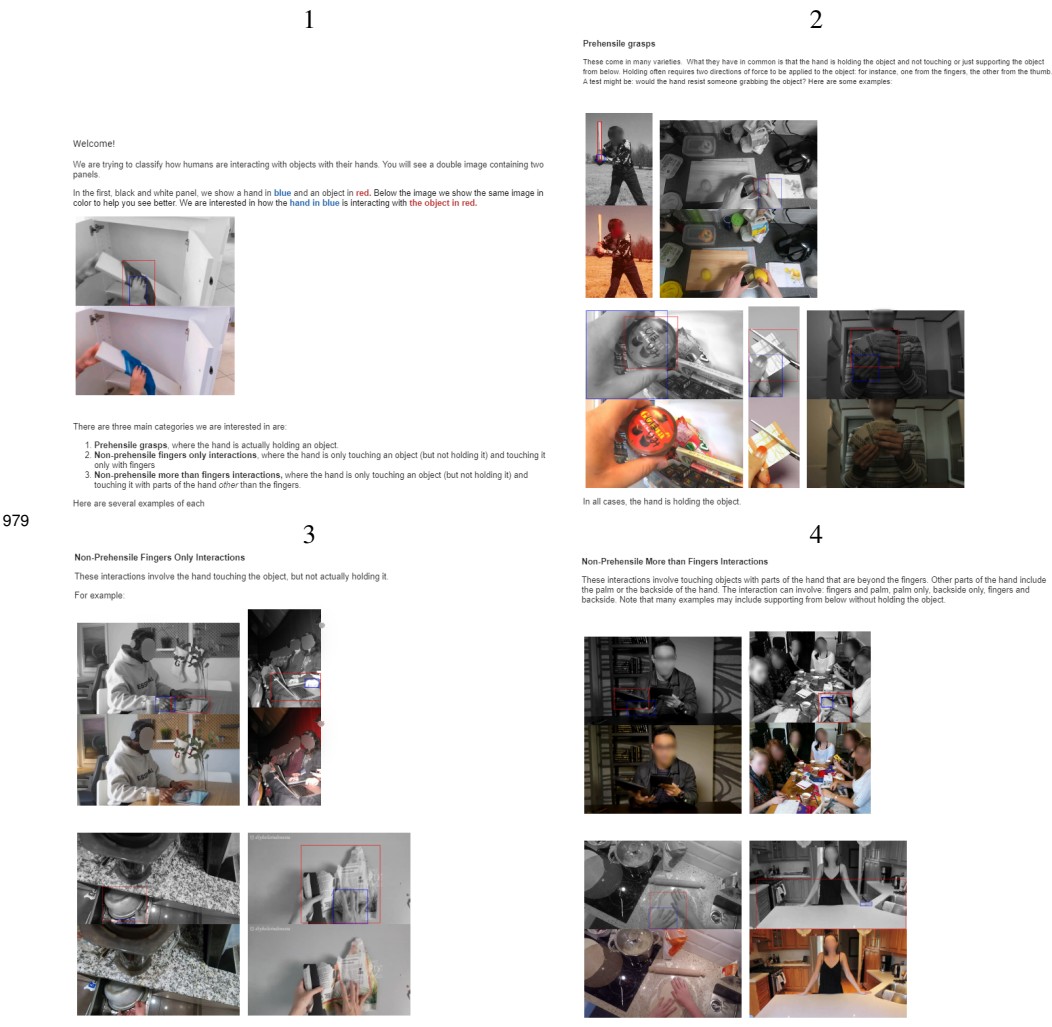

## H.9 Differentiating Prehensile Grasps

We then obtain annotations for different types of prehensile grasps. The instructions were substantially ore complex and the annotation was challenging. We aimed to provide many examples, including illustrations from [12]. Generating fair quality checks was important and we aimed to avoid ambiguous cases.

## 1

Welcome!

Please note: this task requires careful reading. Guessing will not work.

We are trying to classify how humans are interacting with objects with their hands. You will see a double image containing two panels.

In the first, black and white panel, we show a hand in **blue** and an object in **red**. Below the image we show the same image in color to help you see better. We are interested in how the **hand in blue** is interacting with **the object in red**.

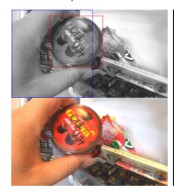

We are looking to categorize human grasps into **five** categories. There are two key concepts we'll explain that together explain the grasps.

- precision and power grasps
- circular and prismatic grasps

**985**

## 2

We are looking to categorize human grasps into **five** categories. There are two key concepts we'll explain that together explain the grasps.

- precision and power grasps
- circular and prismatic grasps

**Precision and Power**

The first distinction is between precision grasps and power grasps. Precision grasps are used for precise movement, and power is used to exert force on an object. They are distinguished by looking at how much contact happens outside the finger tips and whether the fingers can freely move.

|  | Amount of contact not on the fingers | Fingers freely moving? |
|---|---|---|
| Precision Grasp | The hand is using only the finger tips to contact the object (or at least it's almost entirely the finger tips). The object is not in contact with the palm, and almost all of the contact is at the finger tips. | The fingers can freely move to change where the object is. For instance, a precision grasp of a pen lets you move the pen around easily even if your wrist remains still. |
| Power Grasp | Parts of the palm of the hand are in contact with the object or most of the contact is not at the finger tips. For instance, if you use all of the fingers to hold an object. | The fingers cannot freely move. Instead, the object is moved around by moving the wrist. |

**Precision Examples:**

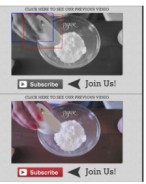

## 3

**Power Examples:**

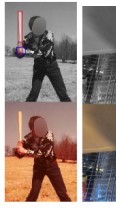 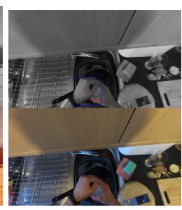 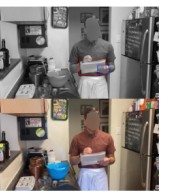

**986**

## 4

**Circular and Prismatic**

The second distinction is between circular grasps and prismatic grasps.

Circular grasps involve finger tips applying pressure on the object from multiple directions. Prismatic grasps involve the tips of fingers 2, 3, 4, and 5 applying pressure from one direction, often in the opposite direction of the thumb. Sometimes three fingers are doing a prismatic grasp and a fourth fingers is doing something. In this case, please mark it as a prismatic grasp.

|  | Shape of the fingers that are in contact | Directions of force of the fingers that are in contact |
|---|---|---|
| Circular Grasp | Fingers form a circle around the object. | The fingers apply force from multiple opposing directions in a circle. |
| Prismatic Grasp | Fingers form a cylinder or line and apply forces from one side. Note that fingers 2, 3, 4, 5 apply forces in one direction. The thumb opposes this direction. For example: | The fingers apply force from one direction. Often the thumb applies force from another direction |

**Circular grasps:**

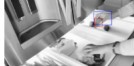 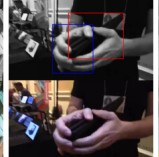 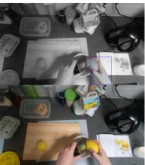

## 5

**Prismatic grasps:**

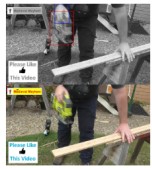 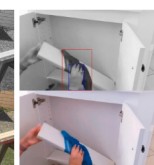 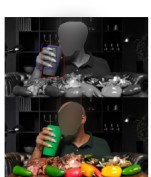

**987**

**Prismatic:** here the hand is mainly doing a prismatic grasp around a knife (the hand is wrapped around it and the fingers are all in one line). However, the index fingers is extended. Please mark this as prismatic.

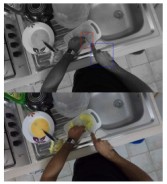

## 6

**Now we will combine them together**

**Power Prismatic** -- This is a classic power prismatic grasp. The palm is in contact (so it is power). The fingers are wrapped around and are not applying force in multiple opposing directions in a circle.

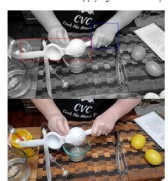

Here are other ones:

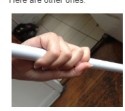 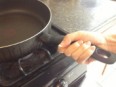

**Precision prismatic** -- here, the fingers are opposed to each other. Just the finger tips are in contact, so it is a precision grasp. Since the in contact fingers are applying force in a single direction and not in a circle around the object, this is a prismatic grasp.

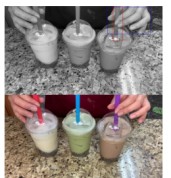

Here are other ones:

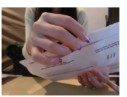

988

**Precision circular** -- this is a classic precision circular grasp. The finger tips are wrapped around the object in a circle and are applying forces in multiple directions. But only the finger tips are in contact.

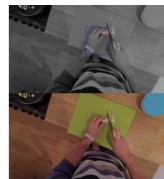

Here are other ones:

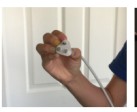
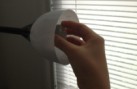

**Power circular** -- the palm and lots of the finger are in contact with the pot and the fingers are wrapped around in a circle.

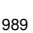

989

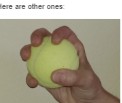
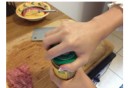

Here are other ones:

**One Exception: Lateral**

Most grasps involve using the insides of the fingers. Sometimes people use the side of their fingers. This is an important exception. This is called a lateral grasp where the object is held between the thumb and the **side** of fingers. This is called a lateral grasp

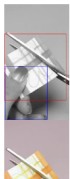

Here is another one:

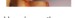
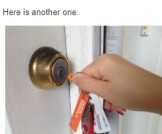

Some common issues

Here are some tricky cases and how to resolve them.

**Grabbing something non-solid**

Sometimes people grab objects that are not solid, such as dough or laundry. **Please mark these as power circular (if the hands roughly form a circle or sphere) or power prismatic (if the hands form a cylinder).**

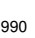

990

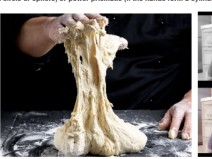
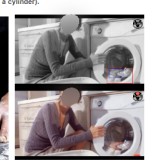

**Grabbing bowls**

We find these cases difficult. You should label these as power circular (if most of the contact is not from the finger tips) or precision circular (if only the finger tips are in contact).

**Grabbing bowls**

We find these cases difficult. You should label these as power circular (if most of the contact is not from the finger tips) or precision circular (if only the finger tips are in contact).

For instance, here, the palm is probably is in contact with the bowl and certainly a lot of the fingers past the finger tip are too.

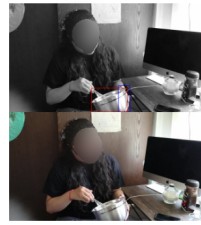

**Funny grasps where fingers are sticking out that don't feel like one category or another**

Many grasps do not fall neatly into these categories. Sometimes the hand has an extra finger touching the object. Please use the closest grasp, or the grasp that you think the hand most resembles. For example, this is a power prismatic grasp, but the thumb is in a weird position. We will try to make the test cases as clear examples as possible.

991

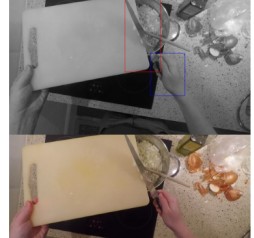

## H.10 Video Identification

Approximately $400 was spent annotating thumbnails to provide training data for automatic relevance filtering. This involved around 21K classification tasks.

## H.11 Filtering Videos

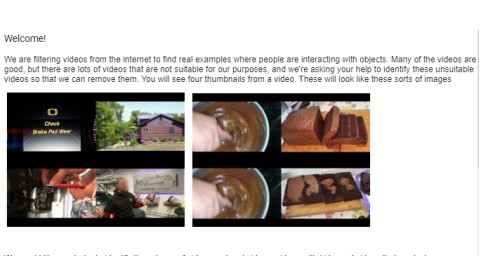

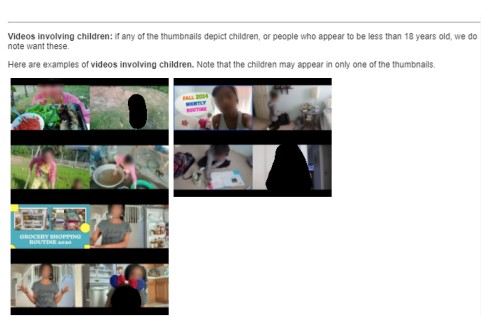

## H.12 Counting Hands

We asked workers to classify videos based on the number of in-contact hands in nine frames from the video.

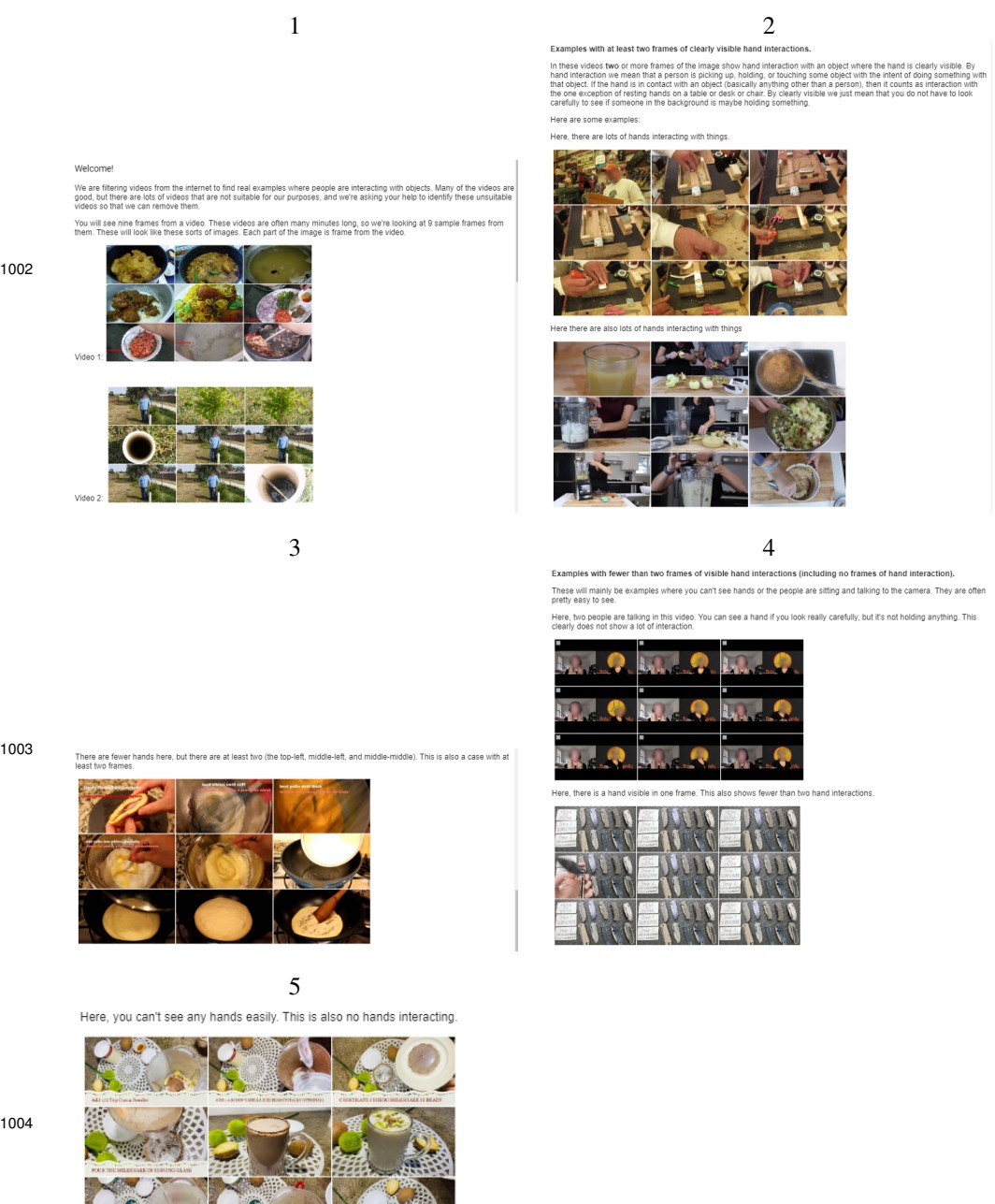

## H.13 Identifying Frame Types

We asked workers to identify up-close vs far-away frames vs non-real images. Gold standard checks were chosen here to be deliberately unambiguous.

1008

Welcome!

We're looking to categorize frames in YouTube videos according to how close to the camera the objects are. We have three main categories based on where most of the objects in the scene are. If there are hands visible, please use the hands that you can see to decide the distance.

**1. Within 0.5m of the camera.**

These are frames where the hands or other objects are very close to the camera and take up much of the frame. These image should feel as if the objects could hit you.

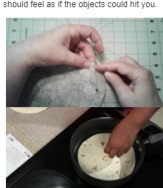
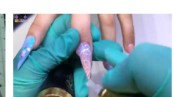

**2. More than 1m away**

These are images where a lot of the scene is visible, and if there are hands, they're probably 1m away. If the object is between 50cm and 1m, please make your best guess as to which is closer. The qualifier and other tests will only have ones where the distance is obvious.

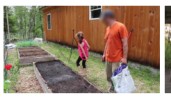
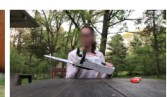
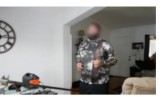

**3. Not a real images**

These are images that are not actually images, since they are things like text or diagrams. If it doesn't look like a real image, please mark it as "not a real image".

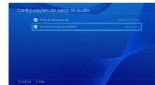

## H.14 Image Redaction

Approximately $4800 was spent redacting children and faces from the dataset, spent over approximately 500K tasks that were primarily binary classification, but also included approximately 17K box annotation tasks.

In this section, we indicate unblurred faces with a black and white checkerboard pattern, and hide minors with a black mask.

### H.14.1 Instructions for Unblurred Face Spotting

1

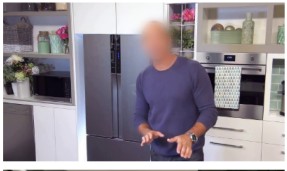

2

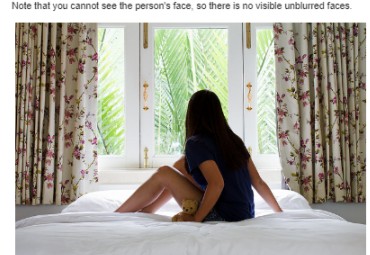

3

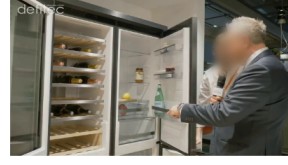

4

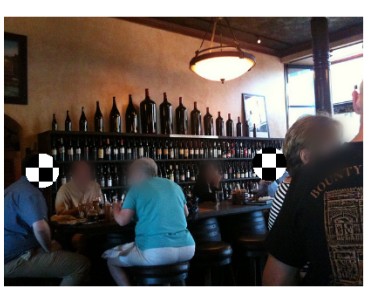

 **H.14.2    Instructions for Unblurred Face Bounding**

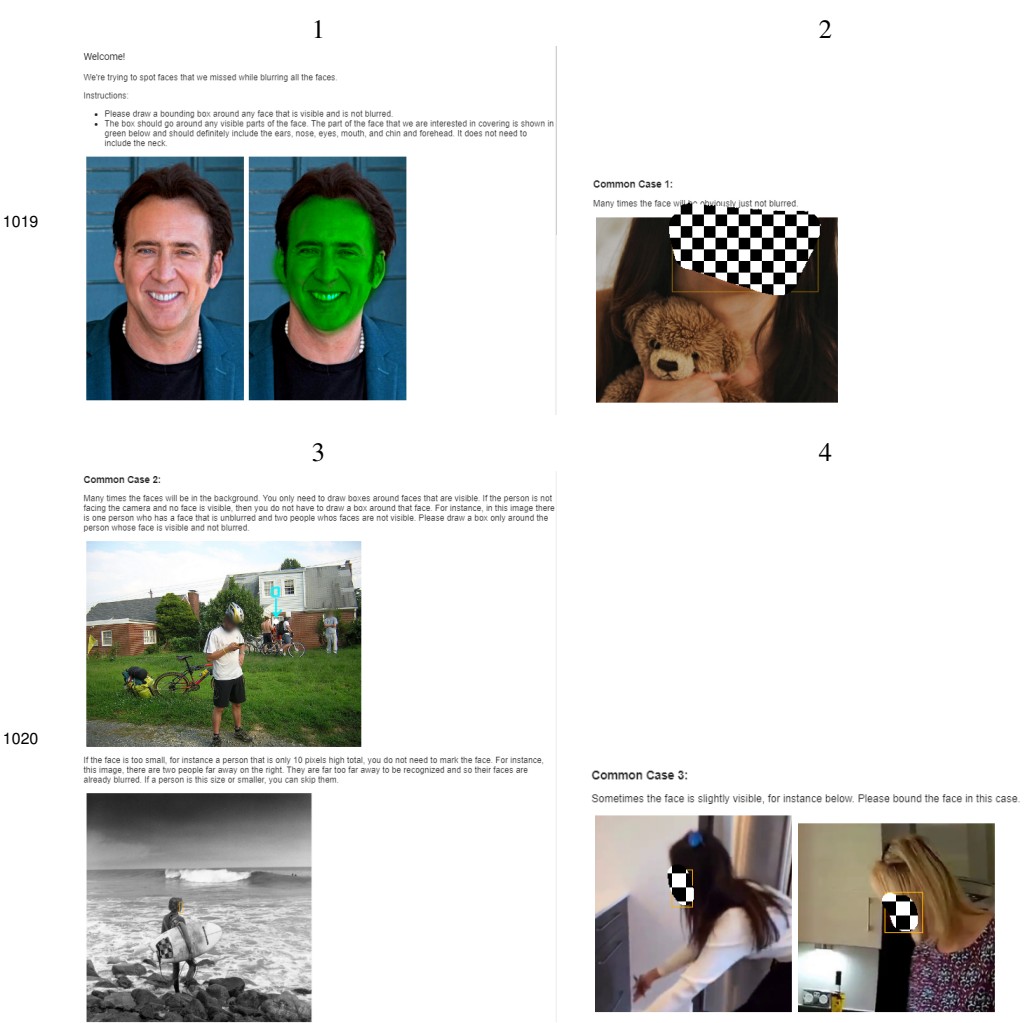

 **H.14.3    Instructions for Spotting Minors**

Workers were instructed to spot children in data and were told that in unsure cases (e.g., someone in their teens), they should mark that person as a child.

1024

**Welcome!**

We are looking to see if there an image contains people who appear to under 18 years of age. The two categories are:

- **Shows children:** it appears that there is at least one child in the image
- **No children:** it appears that there are no children in the image

We are defining children as people who are under 18. In some cases, this may be difficult to tell. If you are not sure, but think it's fairly likely that the person is a child, please mark them as a child. For example, if you're positive the person is 17-19, but not sure where, please mark them as a child.

**Examples of the children category**

Here are examples where there are people who appear to be younger than 18 in the image:

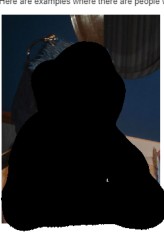



Please note that sometimes children may not have their faces visible. Please label these as children. For example: in this image, the person is clearly a child.

**Examples of the no children category**

Here are examples of no children category

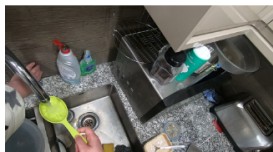

1025

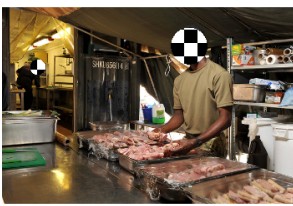

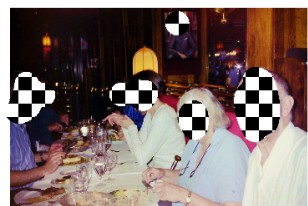

If you cannot recognize whether the person is a child (e.g., a face in a distant crowd that you can't actually see), you can assume the person is an adult. If you can't quite see the person but it seems likely the person is a child (e.g., based on clothing or size), you should mark the person as a child. We will only test you on examples that we believe are clearcut.

## H.15    Polygon Labeling

1027 Approximately $400 was obtaining polygons for comparison with SAM for both hands and objects.
1028 This was done across 2000 tasks. The two tasks are explained below, and follow as similar pattern to
1029 other annotations done in the dataset: a top images illustrates what is to be annotated, and a bottom
1030 image is annotated.

### H.15.1    Hands

1032

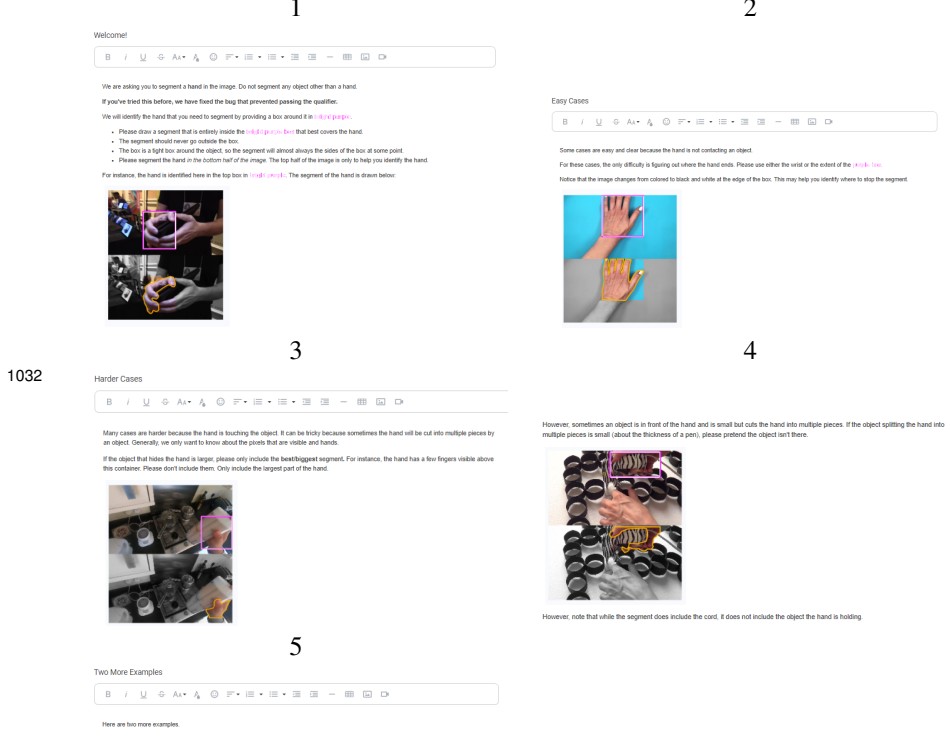

## H.15.2 Objects

1

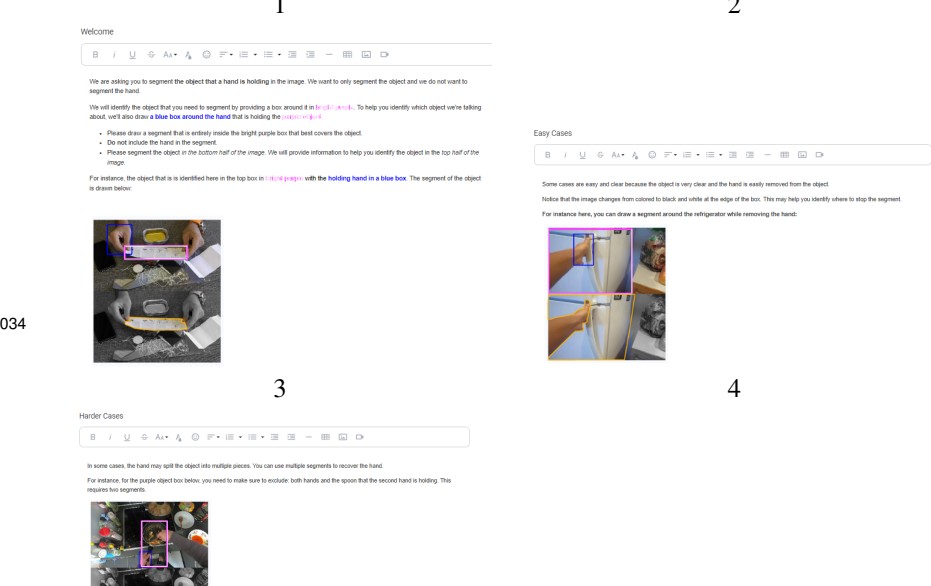

2

3

4