# OpenReview forum: "Towards A Richer 2D Understanding of Hands at Scale"
_NeurIPS.cc/2023/Conference — NeurIPS 2023 poster_

### Official Review · Reviewer_N4EC · 2023-07-04

**Soundness:** 4 excellent
**Presentation:** 4 excellent
**Contribution:** 3 good
**Rating:** 7
**Confidence:** 5

**Summary:**

This paper builds a large-scale hand-object interaction dataset named Hands23 that contains 257K images, 401K hands, 288K objects, and 19K second 33 objects and combines different image sources (Visor, Epic-kitchen, COCO, Internet articulation and novel videos). Compared to the previous hand-object interaction datasets, it includes fine-grained contacts, detailed hand-grasp labels, second objects, and object segmentation. The resulting dataset covers a large amount of hand-object interactions in both ego and exo views. Based on the dataset, they show that a baseline model (Mask-RCNN) can perform well on Hands23 and also generalizes better to other datasets (than previous work). Thus, Hands23 provides a useful resource for the community to study hand-object interactions with applications from computer vision to robotics.


**Strengths:**

1) The proposed dataset is large and covers a wide distribution of hand-object interaction scales in both first and third-person views.

2) Data were richly annotated: hand bounding boxes and segmentation (standard), contact (follows [43], in-contact object (following [53], secondary object (novel), hand grasps (following Cutkosky taxonomy). Thus, the authors innovatively combined to provide a more comprehensive annotation than previous work at scale. Thus, the proposed dataset contains more fine-grained annotations, e.g., second object, detailed contacts and hand-grasp labels.

3). Ethics, privacy and demographics were particularly well addressed. Faces were obfuscated for privacy and Demographics and Realism of the data was analyzed.

4). The proposed baseline model can predict all the rich interaction information and shows good generalization ability to Ego4D dataset.

5) This is a comprehensive paper with excellent Supp. Materials.

**Weaknesses:**

1) One of the main advantages of the proposed dataset is the “second object” defined as an object in contact via a handheld tool. However, the motivation for introducing the second object and the corresponding possible applications are not as well discussed.

2) This paper shows the zero-shot generalization to Ego4D dataset. However, Ego4D only contains the first-person view, which is less challenging due to rel. fixed hand scale and few (hand) occlusions. Therefore, it would be good if the authors could also show the zero-shot generalization to third-person data.

3) While I do think the paper makes significant advances (thus my score), it is not higher, as one would hope for even richer annotations.

4) One more minor comment, the authors state that “We only annotate non-crowd instances with at least 10^3 pixels of area.” While that makes sense it would also be good to annotate crowded areas (just like it is done for pose estimation data in COCO).

Regarding relations to prior work I have two minor suggestions:
- This paper gives fine-grained contacts annotation ({touch, hold} and {tool, container, neither}). However, some previous works in human object interaction (e.g., Discovering Human Interactions with Large-Vocabulary Objects via Query and Multi-Scale Detection) provide even more detailed object and contact classes (e.g., unlocking door). Therefore, the advantage of the current annotation type could be further discussed.
- Line 29: “we extend this to a richer vocabulary that distinguishes touching, holding and using, and includes grasps” Here, GRAB [56] should also be mentioned as it does something similar. Perhaps one should also state that EPIC kitchen has a much broader vocabulary?



**Questions:**

Overall the paper is clear,  but I do have a few suggestions/questions:
-	Object labels in the figures are not too self-evident (e.g., ‘ObjC, NP-Fin’ in Figure 1, etc.)
-	The caption for Table 2 does not mention the metric (the text does, but would be good to add to caption)
-	In the caption of Table 2 you state: “no other dataset produces good results on Hands23.” Yet, 100DOH gives a reasonable 78.5. Likely the only difference is, thus, scale? What if you train on a fraction of Hands23 that has as many images as 100DOH?
- Last but not least, I would find it really useful to get an assessment of the accuracy of the human labelers.

Overall, this is a strong paper, and indeed I was wondering why they did not include Ego4D in the dataset. This comment clarified it: “We were unable to annotate Ego4D due to strict restrictions on data redistribution, but can use self-training to integrate Ego4D and our data” That’s great, but could already be mentioned earlier and I think Ego4D and a few other datasets (e.g. Grab, Discovering Human Interactions with Large-Vocabulary Objects via Query and Multi-Scale Detection) should be in Table 1.

In the Supp Mat it is stated that the data format for the release is not clear yet. I think it would be ideal to follow, e.g., COCO or Epic kitchen for consistency in the literature.

**Limitations:**

---

> ### Author Rebuttal · Authors · 2023-08-09
>
> We thank the reviewer for their detailed feedback which we believe will improve the paper.
>
> **Q1: Could you provide more motivation for second-objects?**
>
> **A1:** We will include a more explicit and centrally organized discussion of second objects if accepted. Since humans use tools to accomplish tasks, second objects are needed for a complete and rich understanding of hand interactions. Consider, for instance, a hand holding a knife. Without the second object, one doesn’t know whether the action is cutting bread, transferring chopped garlic to a pan, scraping something off a cutting board, or even brandishing the knife. Indeed, knowing this distinction is important for connecting to narrations: if one is drilling into a table, the second object is the part that is described as the direct object, rather than the drill.
>
> **Q2 Does the model generalize well on 3rd person data?**
>
> **A2:** Indeed, our motivation for the diversity of data used is so that the model has a good chance of generalizing well to 3rd person data. We have some partial generalization experiments while testing on other datasets as has been noted. These, however, test only hand detection. We will add generalization to other 3rd person datasets if accepted. In the meantime, we show some results on the SAM dataset in the figure pdf. We selected SAM because it is diverse, recent, and high quality. Figure 4 shows two rows of results on SAM; the top is selected and the bottom is random. Our method generalizes fairly well on unseen 3rd person data.
>
> **Q3: What about even richer annotations**
>
> **A3:** We highly welcome suggestions for more labels to further enrich Hands23. We additionally will ensure that is possible for the community to add more annotations.
>
> **Q4: In COCO, why only annotate non-crowded instances with at least $10^3$ pixel area?**
>
> **A4:** We originally did this because we thought that the crowds of COCO (often next to baseball fields or tennis courts) were too small for hands to be reliably understood and parsed. We note that $10^3$ pixels are under $32^2$, which likely leaves each hand being just a pixel. However, we will re-examine this design decision to see if there are many hands amongst crowd instances that are visible.
>
> **Q5: Suggestions about prior work, discussion of Ego4D**
>
> **A5:** Thank you. We will add a discussion about these works and add them to the table. Briefly, we think that our less fine-grained annotations can be obtained very easily and provide a great first initial parse of the interaction. We see these works as complementary, and think the community benefits from having data at different levels of granularity and scale. We will additionally move the discussion of Ego4D and difficulty annotating it earlier in the paper.
>
> **Q6: Paper suggestions**
>
> **A6:** Thank you. Will will fix these if accepted.
>
> **Q7: Is the performance gain of Hands23 primarily due to scale?**
>
> **A7:** Thanks for this great question. To check if scale really matters, we trained the hand detection model using the same amount of data (90K) as the 100DOH trainval set. The resulting model obtains 83.4 AP (Hands23 with 90K images), which is close to the original model’s 85.6 (Hands23 full), and higher than 100DOH’s 78.5 AP. This experiment suggests that not only scale, but also the diversity of the dataset matter.
>
> Furthermore, if one evaluates only on subsets of the dataset, while the 100DOH model often does as well as the Hands23 model, the 100DOH model’s performance is substantially worse on other subsets. For instance, on VISOR, New Videos, Articulation, Hands23 is better by 2.3, 2.3, and 2.6 AP respectively. However, on COCO Hands23 is better by 17.6%. We note that this is only testing hand detection; we have often found that the other aspects of performance struggle (e.g., 100DOH is known to struggle with egocentric hand side as reported on the project website)
>
> **Q8: Will the dataset format match prior work?**
>
> **A8:** Yes, we intend to have the dataset format match prior work so that it is as easy as possible for the community to use.

---

> > ### Comment · Reviewer_N4EC · 2023-08-10
> >
> > Thanks for the clear answers and experiments addressing my comments! I was and remain very positive about this paper.

---

### Official Review · Reviewer_87MP · 2023-07-05

**Soundness:** 4 excellent
**Presentation:** 3 good
**Contribution:** 3 good
**Rating:** 5
**Confidence:** 3

**Summary:**

This paper introduces a hand object interaction-related dataset, Hands23, that providing rich labels for hand images (segemantion and boundingbox of left hand, right hand, object and second object, types of contact, grasp and touch). It labels the existing three datasets (COCO, EPIC-KITCHENS VISOR, and Internet Articulation), as well as self-collected videos. Based on the new dataset with rich labels, the paper uses RCNN to train a multi-task model, which performs and generalizes well.

**Strengths:**

- [Size and rich labels] The presented dataset is much larger than existing datasets and provide segemantion and boundingbox labels of hands and objects.

- [Experiments] The model trained on Hands23 for hand or object detection shows good generalization performance. It outperforms the models trainend on other datasets.

- [Annotation] The annotation procedure is elaborated in great detail in the supplementary.

**Weaknesses:**

- [Multi-task] The paper highlights the importance of rich labels for hands and introduces a multi-task framework (Fig.3). It would be interesting to provide more insights on the interaction of different sub-tasks. For example, do the auxiliary labels affect hand segmentation or bounding boxes? what is the benefits of richer labels?

**Questions:**

- [Action] As Hands23 consists of a large number of videos, what do you think about action labels?

- [SAM] The segmentation labels are generated with the help of Segment Anything (Line.142). I am wondering if the paper could provide some observations or insights when working with Segment Anything or even other large models to generate labels.

**Limitations:**

Yeah. The authors have addressed the limitations in the conclusion section.

---

> ### Author Rebuttal · Authors · 2023-08-09
>
> We thank the reviewer for their thoughtful review and answer the questions below.
>
> **Q1: What are the benefits of multi-task training? Does it hurt the performance on segmentation?**
>
> **A1:** The benefits of the multiple tasks is primarily the richer predictions, which we envision will enable various applications such as robotics, hand-grasp analysis, and parsing video datasets for understanding hands or finding demonstrations.
>
> Once losses are scaled properly, we think that multitask training does not hurt performance much. We also trained a version consisting only of 3-class (hand, first, and second object) instance segmentation without extra heads. The performance was similar. Hands were slightly better (+3.7 detection, +3.5 segmentation), first objects were slightly worse (-1.2 detection and segmentation), and second objects were slightly worse (-1.7 detection, -0.4 segmentation). We believe this variance is within the range of random seeds and the intrinsic minor differences when comparing two setups, but will provide a more detailed analysis if accepted.
>
> **Q2: What do you think about action labels?**
>
> **A2:** We hope that our model will help with action understanding. We do not have action labels for the new videos subset of Hands23, but we chose VISOR precisely since it is connected to the excellent EPIC-KITCHENS action labels. Beyond labeling the dataset, we do believe that the output of our system could help serve for action recognition since the state of hands, first objects and second objects often are informative for action recognition.
>
> **Q3: How does SAM perform on predicting masks?**
>
> **A3:** We will include additional information about SAM if accepted, since we did find a number of interesting observations while both using SAM and trying to develop our SAM-like system.
>
> Briefly, we found that SAM generally performs well and robustly when provided with bounding box prompts. There are some limitations and cases where more failures emerge. One is that SAM has difficulty deciding the end of the wrist for hand bounding boxes and can produce jaggy masks as the wrist. Another is that SAM can have trouble with thin objects where multiple objects appear in the same box. SAM occasionally will segment another object rather than the intended one. However, overall, as shown by our analysis at L238, SAM produces quite good results. These are where failures tend to be more frequent, rather than places where SAM systematically does not work.
>
> We also want to point out Section D in the appendix, where we discuss SAM compared to our own similarly structured system. We compare the proposed model’s performance between using masks from SAM and our own internal mask prediction system. The bounding box detection performance is largely identical, but segmentation is much better using SAM.

---

> > ### Comment · Reviewer_87MP · 2023-08-19
> >
> > Thanks for the rebuttal. The authors solved my concerns. I agree that the paper's novelty is insignificant, but I also appreciate the effort to collect and annotate such large-scale data related to interaction. Thus, I still tend to borderline accept.

---

### Official Review · Reviewer_oZYn · 2023-07-05

**Soundness:** 2 fair
**Presentation:** 3 good
**Contribution:** 2 fair
**Rating:** 4
**Confidence:** 4

**Summary:**

The paper presents a method and dataset for hand-object understanding. Specifically, the dataset provides bounding boxes and segmentation annotations for (1) hands, (2) objects that are in contact with hands, and (3) second objects touched by tools. The annotation also includes contact and grasp type for second objects touched by tools. Overall, there are annotations for ~260K images spanning four different datasets. The segmentations masks are automatically extracted using the off-the-shelf method, Segment Anything.

**Strengths:**

- The proposed dataset has images from both first-person and third-person views. This can help develop hand interaction detection methods that generalize well on different domains.
- The dataset has images from other popular benchmarks like EPIC-KITCHENS and COCO. This could be beneficial to the community to develop and train unified methods for human understanding.
- The authors demonstrate the benefits of data through cross-dataset experiments.
- The proposed dataset annotates richer contact vocabulary, such as distinguishing between touching vs holding.

**Weaknesses:**

The proposed architecture does not convince me fully. There are a few major concerns.

- Currently, the method first detects three 'object' classes: hand, object, and second object. These classes are detected independently. However, there is a clear correlation between these three classes. For instance, the object's location in contact with the hand depends on the location of the hand. Also, the location of the second object depends on the location of both the hand and the first object. However, in the current method, this is not reflected; the three classes are detected independently.
- In L217-L218, every hand is matched to an object with the highest interaction score. This excludes the possibility that a hand can contact more than one object. For example, a hand can hold a knife and the chopping board at the same time. The current model design does not reflect this.
- When detecting object and second-object interaction, does the model also consider hand-object interaction? For example, consider object A (first object) and object B (second object). When inferring the interaction score between the two objects, will the model consider if object A was in contact with the hand? If object A was not in contact with the hand, then object B cannot be the second interaction object. Is there any such explicit modeling?

**Questions:**

Please see the weakness section.

**Limitations:**

The authors adequately addressed the limitations and potential negative societal impact of their work.

---

> ### Author Rebuttal · Authors · 2023-08-09
>
> We thank the reviewer for their thoughtful review of the paper and respond to their three questions and weaknesses below.
>
> **Q1: Does the method model the correlation between three classes?**
>
> **A1:** Thank you for the great question. It is true that our model treats the three classes independently in terms of formulation, but by using a shared backbone, we believe that the model is capable of internally and implicitly modeling the relationships between the three classes. We choose to frame the problem as independent to make the problem amenable to existing object detection machinery. Moreover, the relationship is detected based on features from both objects as well as the relative location of paired bounding boxes (more details at line 211).
>
> To verify our hypothesis of the implicit modeling of object relationships, we provide qualitative results from videos in Figure 2 showing all detected objects above a low threshold (0.8 for hands, 0.3 for first and second objects) without association inference. The *before* image shows the object without the hand and the *after* image shows the hand in contact. In the before example, the object is not detected or detected with a lower confidence score. In the after example, when the hand appears, the object is detected or its score increases. This suggests that the model is implicitly using the other object categories during its detection.
>
> **Q2: Can the model capture cases of one hand in contact with multiple objects?**
>
> **A2:** Thank you for the thoughtful comment about cases of a hand in contact with multiple objects. In data annotation, we annotated a single hand/object contact to make data annotation scalable. However, our model is, in principle, able to handle the multiple contact case because it predicts contact state for each pair of hand and object as an independent classification (i.e., the objects do not compete in the prediction). This is unlike the Shan et al. formulation, which explicitly precludes multiple contacts by using a single offset vector.
>
> If accepted, we will include a more thorough analysis and investigate additional annotations. In the meantime, we did a small experiment to see if the model has this capacity (since during training, it is identifying which object is annotated as in contact). Instead of keeping only the object with the highest in-contact score, we modify the inference algorithm to keep all predicted in-contact objects with a high contact score (threshold=0.8) and low IoU with other in-contact objects (IoU threshold = 0.2). In Fig 3, we show examples of multiple-object contact: (a) a hand holding a brush and touching the surface of a phone case, (b) a hand grabbing food and putting them into the bowl, (c) a hand holding two beans, and (d) a hand holding the comb and a strand of hair. In other cases like (e), the model will find the object by two bounding boxes because the object is separated by occlusion. There are some failure cases too.
>
> **Q3: Does the model consider hand and first-object interaction before deciding on first-object and second-object interaction?**
>
> **A3:** The inference algorithm does indeed consider interactions between the hand and first-object when detecting interactions between the first and second object. In our inference, the contact association is always starting from a hand. The inference algorithm will only consider first-object and second-object contact if the first object is in contact with a hand.

---

> > ### Comment · Reviewer_oZYn · 2023-08-16
> >
> > I thank the authors for clarifying my questions. I am satisfied with the authors' responses regarding **Q2** and **Q3**. However, I am still concerned regarding **Q1**. The proposed method should explicitly model the correlation between the three object classes. Currently, the three classes are detected independently. Although the proposed method uses a shared backbone, the backbone is primarily used as a feature extractor and therefore does not model relationships between three classes. Given that the task is to detect hands, an object in contact with the hand, and a second object in contact with the first object, the proposed method should model such relationships. However, I do not see any such formulation here. Therefore, I also agree with Reviewer ufdR's concerns regarding limited technical contribution and keep my original rating borderline reject.

---

> > > ### Author Response · Authors · 2023-08-21
> > >
> > > We are glad that our responses to Q2 and Q3 resolved the previous concerns. We thank the reviewer for additional feedback that can let us more directly discuss the reviewer’s concerns about explicit and implicit modeling.
> > >
> > > **Is there any explicit modeling in the framework?**
> > >
> > > Although the detection framework models the detections independently, we do want to point out that the association component (and thus the full detection system) does explicitly model interactions between hands, first objects, and second objects. In particular, the network is not only looking at top detections to find first objects and second objects, but instead, the contact association MLPs model the in-contact relationships. In the inference process, a hand must be detected first; if the hand is predicted as object-contact with a particular object, only then is that object produced as a detection. Similarly, a first object must exist in order for a second object to be associated with it and therefore detected.
> > >
> > > **Is the backbone just feature extraction?**
> > >
> > > We understand the reviewer’s comment about the backbone being just feature extraction, and this is certainly true for many detection models. However, in our case, we would point out two particular details that, in our view, make the backbone not a generic feature extractor but instead the part of the network that does most of the work:
> > >
> > > - First, the backbone is a 101-layer deep ResNeXt-101. By the time the data has reached the detection heads, most of the computational work and processing has been done – the MLPs are a handful of layers and depend on the backbone having already understood the object categories.
> > >
> > > - Second, while the backbone is ImageNet pretrained, it has been further trained for 400K iterations with a batch size of 16, thus seeing >6M training images (and thus seeing ~6B proposals), and therefore being shaped by the detection task and the auxiliary MLP heads. To correctly recognize that an object is a second object, the network needs to implicitly spot hands and a tool in use. If the network generically finds potential second objects (as opposed to objects that are currently contacted by a tool), it incurs a high loss because these are not labeled as second objects. Indeed, our figures in the rebuttal suggest that the backbone is doing this.
> > > With a sufficiently deep network, we believe that such modeling is possible, much like how GPT-4 is able to model fairly complex grammar structures implicitly through sufficient complexity.
> > >
> > > **What’s the technical contribution?**
> > >
> > > In our view, the strongest technical contribution of this paper (as well as many other papers) is not in algorithmic or generic ML machinery, but rather in problem formulation, extensive work on dataset creation, and analysis. Please see our response to ufdR for a more extensive discussion.

---

### Official Review · Reviewer_ufdR · 2023-07-07

**Soundness:** 3 good
**Presentation:** 3 good
**Contribution:** 1 poor
**Rating:** 3
**Confidence:** 4

**Summary:**

This work presents a framework that outputs rich interaction information for hands that are in an interactive state on a daily basis, and a large dataset that supports the model. The framework is based on the standard RCNN object detection mechanism and has a simple structure that can be easily extended to meet subsequent information additions for interactive hand understanding.

**Strengths:**

- A large dataset of human hand interactions is proposed, which provides rich and high-quality data annotation that facilitates the understanding and prediction of human hand behavior.
- A framework for manual interaction understanding, with a simple structure that is easy to extend and conducive to directly supplement more interactive information.
- Additional categories of how human hands interact with objects are classified and understood.

**Weaknesses:**

- The understanding of human-object interaction is still in the form of classification mapping, which is a large and tedious classification and lacks simplicity.
- The framework is based on the object detection approach, which may be less applicable for large scenes or scenes that are messy.
- It could be ragarded as an incremental supplement for the work of Shan et al, not innovative enough.（ Dandan Shan, Jiaqi Geng, Michelle Shu, and David F Fouhey. Understanding human hands in contact at internet scale. In Proceedings of the IEEE/CVF Conference on Computer Vision and Pattern Recognition, 482 pages 9869–9878, 2020.)

**Questions:**

The narrative of the article is clear and there is little doubt.

**Limitations:**

A large number of 3D hand-object interaction state estimation methods already exist in the community, which provide a more fine-grained analysis of contact. The prominent problem with current 2D-level interactions exists in the robustness to small and multiple targets, but these are subsumed into the Limitation of this work.

---

> ### Author Rebuttal · Authors · 2023-08-09
>
> We thank the reviewer for their review. We respectfully disagree on a number of key points that we will describe below.
> We believe that the reviewer is suggesting that 3D hand-object reconstruction is sufficient to solve the task we tackle, and so we will respond to this first before responding to the three weaknesses.
>
> **Q1: Can 3D hand-object reconstruction solve hand-object interaction understanding?**
>
> 3D hand-object reconstruction does provide an alternate fine-grained characterization but often does not generalize well to realistic data such as internet videos, and often only works on a fixed set of objects with given 3D object models or a limited set of (often hand-held) objects. In the real world, many objects that can be in contact with hands. Some objects like large fridges and tables may be larger than the full frame and be occluded and impossible to reconstruct. Others may be tiny and too numerous to have reliable reconstructions, such as the many small pieces of hardware while assembling furniture. Sometimes hands have self-contact with the body. Further, tools interact with objects as well, which is not captured by existing hand-object reconstruction methods, to the best of our knowledge.
>
> Our method finds this interaction in realistic data, possibly with many people and without needing a fixed set of objects in advance. In fact, 2D models such as the ones we use have been used to localize hands and objects for 3D methods, such as Frankmocap which uses the Shan detector to work on isolated hands. This is, of course, in addition to a wide variety of other tasks including action recognition, interactive object understanding, and robotics manipulation learning.
>
> As a demonstration of the difficulty of the data, we show two existing interaction models on the hands23 test set in Figure 1 in the figure response. As shown, none of the methods generalize well.
>
> **W1: The interaction formulation is large and tedious and lacks simplicity**
>
> **A1:** We respectfully disagree with this characterization. The method is simple enough to fit into one page, and we provide a few extra classes that are directly usable, as shown by the application examples such as Figure 6 and supplement section A.3. These analyses let us understand the diverse/rich types of hand/object/second-object interaction that exist. We genuinely do not understand what is meant by tedious. If the reviewer is referring to the pairwise interaction classification, we note that this is simple and can likely be further accelerated with various tricks.
>
> **W2: Are detection-based frameworks less applicable to large and messy scenes?**
>
> **A2:** We respectfully disagree. We precisely chose to formulate the problem as object detection/instance segmentation because we want to tackle large, messy scenes with unknown numbers of objects. Indeed, we show results on COCO, which has many scenes with multiple people and in fact, this is where we show a large improvement over Shan et al. in hand association (74.2 vs 55.9 F1 for 6+ hands). Detection is the standard approach for handling an unknown number of objects, and without a concrete alternative, we’re at a loss to respond.
>
> **W3: “It could be ragarded as an incremental supplement for the work of Shan et al, not innovative enough.”**
>
> **A3**: We strongly disagree with this characterization. The paper introduces (a) a richer set space of labels, including first and second object contact which is not provided at this scale in any other dataset, as well as grasps and fine-grained contact annotations; (b) annotations for 250K images with clear copyright status and data protections; (c) a method to produce the rich set of labels that improves over Shan et al., especially for complex scenes; (d) various smaller contributions such as SAM-enabled segmentations, analysis of potential biases in detection performance. It is true that there are similarities to Shan et al., which is because the core idea works well and has seen substantial use in many downstream tasks. Building on this past work is a strength, not a weakness.

---

> > ### Comment · Reviewer_ufdR · 2023-08-16
> >
> > Thanks for the author's response. From their reply, it seems that the authors are rather dissatisfied and believe that I might not have fully comprehended or appreciated their work.
> >
> > The core idea of this paper is that the authors believe that it's plausible to model the interaction state of hands and objects using bounding box associations, and even extend this concept to predicting future interactions involving a second object. However, in my perspective, such a model might not be effectively suitable for complex scenarios. This viewpoint is echoed to some extent by the authors themselves in the "Limitations" section, particularly when bounding box regression errors are prominent. It's not that I dismiss the potential of a simple model outright, but rather due to the oversimplified nature of bounding box descriptions (or the extracted features) that completely discard crucial information such as hand poses – which is the underlying reason behind my inquiry about 3D hand poses.
> >
> > The model and methodology proposed by the authors are akin to attempting to discern intricate details of an object from a heavily blurred image. It's not that it's inherently impossible, but rather that it defies conventional wisdom. Of course, I speculate that we could make educated guesses aided by large datasets:-)
> >
> > While I do acknowledge the commendable objectives the authors aim to achieve, after thoroughly considering their response, my conviction remains unchanged: I still think this paper lacks **technical contributions**. In fact, the use of bounding boxes to represent interaction states has already been explored in previous work, and the attempt in this paper to introduce second-object prediction doesn't significantly alter this paradigm.
> >
> > The most contribution of this paper, it seems, is the dataset. Nonetheless, I remain unconvinced that this in itself meets the bar of NeurIPS.
> >
> > Consequently, regarding this work, I do not change my score as reject.

---

> > > ### Author Response · Authors · 2023-08-21
> > > **Official Comment by Authors (1/2)**
> > >
> > > We thank the reviewer for the detailed description of their concerns. We appreciate these additional details since they have helped us better understand the concerns about the paper and the reviewer’s perspective. Since the discussion period is short, we understand that we likely cannot have another back-and-forth, but do want to provide a response for the reviewer before the period ends.
> > >
> > > We’ve attempted to summarize the concerns into questions to help explain what we believe the concern is and to organize the discussion. We apologize if we misinterpret and do not mean to speak on the reviewer’s behalf.
> > >
> > > **Q1: Does it make sense to model hand-object interaction using 2D bounding boxes, especially in complex scenarios, and compared to hand poses and other such representations?**
> > >
> > > In short, we think that reliable 2D information about hands and objects can have a substantial impact, and we believe that the value of 2D bounding boxes can be seen by their use in downstream applications.
> > >
> > > While it may defy conventional wisdom for some in the community, we do believe that others in the vision community and further afield find the output valuable. To get a sense, we looked through papers that cite the Shan et al. paper. In the past week, two Arxiv papers  [A,B] have directly used the Shan et al. model in downstream tasks: using all of its boxes [A] and using it as a way of finding the object in contact when a semantic detector cannot find an object mentioned in a caption [B]. Indeed, researchers in a wide variety of fields have used the Shan et al. model, for instance in robotics [31,60,68], action understanding [15,20], and post-stroke rehabilitation [58]. By providing a fuller understanding of contact via second objects (see Q1 for N4EC for a succinct summary of why the second objects are needed), we think our proposed dataset, labels, tasks, and model can have even more impact.
> > >
> > > The reason why these downstream tasks use the box model is that boxes are often sufficient for tasks like video pretraining, tracking held objects, finding a held object for reconstruction, etc. Indeed, because box-based systems only produce a box, such systems can work reliably on the wide variety of objects that will appear in daily life and in a wide variety of images, even without a 3D model and when the object is hard to name (the use case for [B]). While 3D hand pose has become far more reliable, hands can be collected in great quantities and are a single (albeit deformable) object for which a basis can be learned. In contrast, hand-held objects and tool-contacted objects are far more varied, and suffer from a long tail.
> > >
> > > We agree with the reviewer that in the long-run, it is good to have a more detailed analysis of hands, objects, and interactions such as 3D hand poses, contact (such as what is being captured in-lab), and forces. At the same time, we think that multiple levels of richness provide value: SMPL does not make pedestrian detection obsolete since one may be able to spot a pedestrian reliably even if the reconstruction would be too difficult.
> > >
> > > In order to get to the richer representations that we all want, we see boxes as a necessary first stage. For hand reconstruction, the first thing that needs to be known is where the hand is, and whether it is right-vs-left. The proposed system provides this, plus auxiliary information that may help with understanding or categorizing interaction. We likewise hope that future reconstruction systems will be able to parse the rich 3D interaction shown on the left of Figure 1 of the paper. However, we suspect that such systems will first need a detection system to provide where the objects are and what is connected with what (for instance, to determine which contact losses to impose in reconstruction). Providing such detection information reliably is itself a challenge and requires data, labels, and models.

---

> > > > ### Author Response · Authors · 2023-08-21
> > > > **Official Comment by Authors (2/2)**
> > > >
> > > > **Q2: What are the technical contributions of the paper, especially considering that other systems have shown bounding box-based analysis?**
> > > >
> > > > We now believe we understand this concern – we believe the reviewer is asking: given that there is an existing 2D detection system for hands, what is the contribution of the new system?
> > > >
> > > > We agree with the reviewer that the paper’s contribution is not in algorithmic novelty, such as a different loss function or neural building block that is demonstrated in multiple systems. We do think that the improved association is important; show it improves results in complex scenes (+18.3 association F1 in scenes with >=6 hands); and think it is more simple than the Shan et al. association method. However, we concur with the reviewer that the algorithmic contribution is itself, not enough for a NeurIPS paper.
> > > >
> > > > We think the technical contributions lie elsewhere, like in many data-driven papers. While contributions can be algorithmic or model-based, we think that innovation also comes from data, labels, and tasks. In particular, we’d order the contributions of the paper (with a particular focus on contributions over Shan et al.) as:
> > > >
> > > > (1) the dataset and labels, including substantial effort to cover the full space of possible input settings, as well as clear copyright status (note that Shan et al. does not guarantee CC copyrights or contain egocentric data) and privacy efforts.
> > > >
> > > > (2) the additional tasks of second objects (which were not available at this scale in any previous work) plus grasp categories and more fine-grained contact information.
> > > >
> > > > (3) the method, including innovations that lead to substantial improvements over Shan et al. for performing associations between hands and objects, especially in complex scenes.
> > > >
> > > > (4) various smaller contributions: SAM-integration and analysis (since Shan et al. only does boxes), including analysis of SAM; analysis of biases in detection performance; and analysis of performance losses due to privacy protections. These are smaller, but collectively have impact.
> > > >
> > > > [A] Helping Hands: An Object-Aware Ego-Centric Video Recognition Model. Chuhan Zhang,  Ankush Gupta, Andrew Zisserman.
> > > >
> > > > [B] Leveraging Next-Active Objects for Context-Aware Anticipation in Egocentric Videos. Sanket Thakur, Cigdem Beyan, Pietro Morerio, Vittorio Murino, and Alessio Del Bue.

---

### Author Rebuttal · Authors · 2023-08-09

We put all figures mentioned for each response in this PDF.

---

### Decision · Program_Chairs · 2023-09-21

**Decision:**

Accept (poster)

**Comment:**

The reviews on this paper are mixed.  Two of the reviewers recommend acceptance, while two others recommend rejection.  One weakness, as pointed out by several reviewers, relate to the ambiguity in the definitions and formulation of the interacting and second interacting objects.

Having carefully read the paper, reviews, and author responses, the AC recommends accepting the paper.  The scale and collection efforts of the paper are impressive and will no doubt benefit the community working on downstream tasks that will use the hand detector.